Upscaling of soil methane fluxes from topographic attributes derived from a digital elevation model in a cold temperate mountain forest

Sumonta Kumar Paul[1], Keisuke Yuasa[1], Masako Dannoura[1], Daniel Epron[1]

[1] Graduate School of Agriculture, Kyoto University, 606-8502, Japan

*Correspondence to*: Sumonta Kumar Paul (paulsumonta@gmail.com), Daniel Epron (daniel.epron.3a@kyoto-u.ac.jp)

ORCID ID

SKP: 0009-0000-7293-7255

MD: 0000-0003-0389-871X

DE: 0000-0001-9451-3437

**Abstract.** Forest soils are generally considered a sink for atmospheric methane ($CH_4$), but their uptake rate can vary considerably in space and time. This study investigated the role of topography and vegetation on soil $CH_4$ fluxes and the temporal patterns of spatially upscaled soil $CH_4$ fluxes in a topographically complex cold-temperate mountain forest in central Japan. We measured soil $CH_4$ fluxes nine times during the snow-free season at multiple locations within a 40-ha area in a forested watershed. Non-waterlogged soils were a sink of $CH_4$, while small wetland patches emitted $CH_4$ consistently throughout the study period. We used a machine-learning approach to upscale the measured soil $CH_4$ fluxes to the landscape scale for non-waterlogged soils at each date of measurement, using topographic and vegetation attributes derived from a digital elevation model and aerial images. The accuracy of predicted fluxes varied seasonally, with the highest model performance observed in early autumn ($R^2 = 0.67$) and the lowest in mid-summer ($R^2 = 0.31$). Predicted $CH_4$ fluxes varied significantly across topographic positions, with greater uptake on ridges and slopes than on the plain and foot slopes. Topography played a predominant role compared to vegetation in the spatial variability of $CH_4$ fluxes. Predicted $CH_4$ fluxes at the landscape scale in the non-waterlogged area ranged from -0.34 to -0.60 g $CH_4$ ha$^{-1}$ h$^{-1}$ in spring, -0.39 to -1.28 g $CH_4$ ha$^{-1}$ h$^{-1}$ in summer, and -0.48 to -0.89 g $CH_4$ ha$^{-1}$ h$^{-1}$ in autumn. Seasonal fluxes were highly correlated with the 20-day antecedent precipitation index ($R^2 = 0.70$), revealing the importance of seasonal moisture conditions in regulating $CH_4$ flux dynamics. This study highlighted the importance of topography in controlling soil $CH_4$ fluxes and the efficiency of remote sensing and machine learning approaches to scale field measurements to the landscape level, enabling visualization of spatial patterns of fluxes across the landscape over time, despite high uncertainty on some measurement dates, particularly for low elevation pixels.

## 1 Introduction

Methane ($CH_4$), the second most important anthropogenic greenhouse gas, contributes substantially to the anthropogenic radiative forcing and is responsible for approximately 0.5°C of current global warming compared to 1850 - 1900 (IPCC, 2023). Natural wetlands (149 Tg $CH_4$ yr$^{-1}$) and rice cultivation (30 Tg $CH_4$ yr$^{-1}$) are important sources of $CH_4$; in contrast, non-waterlogged soils are considered a biological sink of atmospheric $CH_4$, with an estimated uptake of 25-45 Tg yr$^{-1}$, contributing 5-7% to the global $CH_4$ sink (Saunois et al., 2020). Forest soils account for approximately 60% of global soil $CH_4$ uptake (Dutaur and Verchot, 2007), and soil uptake rates are particularly high in Japanese mountain forests due to their high porosity (Ishizuka et al., 2000). $CH_4$ uptake by forest soils is driven by methane-oxidizing bacteria in oxic soil layers, whereas anaerobic environments such as wetland soils are usually dominated by methanogenic archaea producing $CH_4$ (Christiansen et al., 2016). $CH_4$ production can also occur in non-waterlogged soils, either in deeper soil layers or in microsites located in otherwise well-aerated soil layers, if anaerobic conditions prevail (Angel et al., 2012). Hence, $CH_4$ oxidation and production can occur simultaneously at the same location, contributing to the net flux.

Net soil $CH_4$ fluxes depend mainly on the soil air-filled porosity (AFP), which in turn depends on total porosity and soil water content. A high AFP enhances gas diffusion in soil and, consequently, microbial $CH_4$ oxidation (Kruse et al., 1996). Soil organic matter at the soil surface can act as a physical barrier to atmospheric $CH_4$ diffusion and reduce the $CH_4$ uptake rate (Yu et al.,

2017). Conversely, carbon substrates released by the decomposition of soil organic matter can increase $CH_4$ oxidation activity either by directly stimulating the growth of methanotrophs or by promoting $CH_4$ production in anaerobic microsites and indirectly supporting the growth of methanotrophs (West and Schmidt, 1999). Additionally, soil nutrients can influence soil $CH_4$ fluxes by regulating the soil microbial community. The activity of methanotrophic microorganisms is affected by the availability of inorganic nitrogen (Bodelier and Laanbroek, 2004). Although methanotrophic activity can be nitrogen-limited in forest soils (Veldkamp et al., 2013), increasing ammonium ($NH_4^+$) concentration often reduces $CH_4$ uptake due to competitive inhibition by $NH_4^+$ of the enzyme methane mono-oxygenase, which can oxidize both $CH_4$ and $NH_4^+$. Nitrate ($NO_3^-$) can also be a potent inhibitor of $CH_4$ oxidation in some soils (Mochizuki et al., 2012). Although temperature affects microbial activities, including methanogenesis and methanotrophy (Luo et al., 2013; Praeg et al., 2017), $CH_4$ uptake is generally less sensitive to changes in soil temperature than in soil moisture (Epron et al., 2016).

Topography and vegetation cover can create a predictable distribution of soil moisture and nutrients across topographically complex landscapes (Jeong et al., 2017; Murphy et al., 2011). In Japan, forests cover 68% of the land, mostly in mountain areas. Conifers account for 44% of the total forest area (Lundbäck et al., 2021; Nakamura and Krestov, 2005). Topography is a critical determinant of soil hydrological conditions, from well-drained slopes to waterlogged riparian areas (Kaiser et al., 2018). Topography can also impact soil nutrient availability by altering leaf litter accumulation and the movement of soil nutrients (Osborne et al., 2017; Tateno and Takeda, 2003). The spatial distribution of trees, differences in species abundance across the landscape, and variation in litter chemistry often create heterogeneity in soil nitrogen cycling (Osborne et al., 2017). Furthermore, differences in stem flow and throughfall related to differences in canopy structure between tree species can indirectly influence spatial patterns of soil moisture (Holwerda et al., 2006).

*In situ* chamber measurements have long been the dominant method for studying $CH_4$ fluxes in forests, providing insight into the processes that drive them (Brumme and Borken, 1999; Guckland et al., 2009; Itoh et al., 2009). Until recently, most studies reported spatially average flux values measured at several locations (Gomez et al., 2016; Itoh et al., 2009). This method is acceptable for small patches of homogeneous landscapes, such as crops or single-species tree plantations in flat terrain. However, it is inappropriate for more complex landscapes, as the number of sampling points required to obtain an accurate spatially-averaged flux would increase considerably.

In complex terrains, measurement locations can be grouped into several distinct categories according to landforms (Courtois et al., 2018; Gomez et al., 2016; Itoh et al., 2009; Kagotani et al., 2001; Kaiser et al., 2018; Warner et al., 2018), soil microtopographic features (Epron et al., 2016), vegetation characteristics (Guckland et al., 2009), or land uses (Jacinthe et al., 2015). However, as Vainio et al. (2021) pointed out, aggregation assumes spatial homogeneity of fluxes within each category or requires a large number of sampling points to capture the spatial heterogeneity, and this approach ignores the spatially continuous nature of soil processes and their drivers.

More recently, regressions with multiple landscape attributes derived from remote sensing-based maps have been successfully applied to upscale $CH_4$ to a catchment scale (Kaiser et al., 2018). Recent studies conducted on a 12-ha forested watershed (Warner et al., 2019), a 10-ha boreal forest plot (Vainio et al., 2021) , two northern peatland-forest-mosaic catchments of 450 ha and 790 ha, respectively (Räsänen et al., 2021), and a 450-ha subarctic tundra (Virkkala et al., 2024) have demonstrated the effectiveness of machine-learning modelling approaches for upscaling $CH_4$ fluxes from remote sensing data.

Soil $CH_4$ fluxes exhibit strong spatiotemporal variability in temperate mountain forests, and robust large-scale estimates remain scarce despite their importance for consolidating the global methane budget because upscaling fine-scale chamber-measured $CH_4$ fluxes requires an explicit understanding of their spatial and seasonal heterogeneity. We assessed the role of terrain attributes (topography, vegetation) on methane fluxes throughout the snow-free season in a topographically complex mountain landscape, and how the spatial heterogeneity of predicted fluxes and the aggregated fluxes at the landscape level vary over time. We measured soil $CH_4$ fluxes several times during the snow-free season at multiple locations within a 40-ha area in a forested watershed. We applied a random forest machine-learning approach in combination with terrain attributes from

remotely sensed data, i.e., a digital elevation model (DEM) and a vegetation map derived from aerial images, to upscale measured soil $CH_4$ fluxes to the landscape level. We hypothesized that (1) terrain attributes related to water accumulation are reliable predictors of soil $CH_4$ fluxes, (2) spatial patterns of uncertainties in predicted soil $CH_4$ fluxes vary seasonally due to a wet early summer influenced by the East Asian monsoon, (3) predicted soil $CH_4$ fluxes vary within the landscape depending on topography and vegetation, and (4) seasonal variations of $CH_4$ flux at the landscape scale are explained by recent past precipitations.

## 2 Materials and methods

### 2.1 Description of the study site and experimental design

This study was conducted in the forested upper Yura River watershed (520 ha, 35.34 N; 135.76 E) located at the Ashiu Experimental Forest of Kyoto University in northeastern Kyoto Prefecture, Japan (Fig. 1). The mean annual temperature and precipitation were 10.3°C and 2,732 mm, respectively, between 2011 and 2020 and the ground was covered by snow (2-3 m depth) from mid-December to mid-April (Epron et al., 2023). The study area is characterized by a cool-temperate monsoon climate, with a very humid early summer (520 mm in June and July on average between 2011 and 2020) and occasionally heavy precipitation caused by typhoons in late summer. The soils in the study area are classified as brown forest soils according to the Classification of Forest Soils in Japan (cambisols according to the FAO classification), with a relatively thick brownish-black A horizon with a crumb structure and a brown B horizon with a blocky structure (Hirai et al., 1988; Ueda et al., 1993). The forest is primarily dominated by *Cryptomeria japonica* D. Don (Japanese cedar, 73% of the basal area in four 1-ha census plots), mixed with more than 50 broadleaved species (Ishihara et al., 2011).

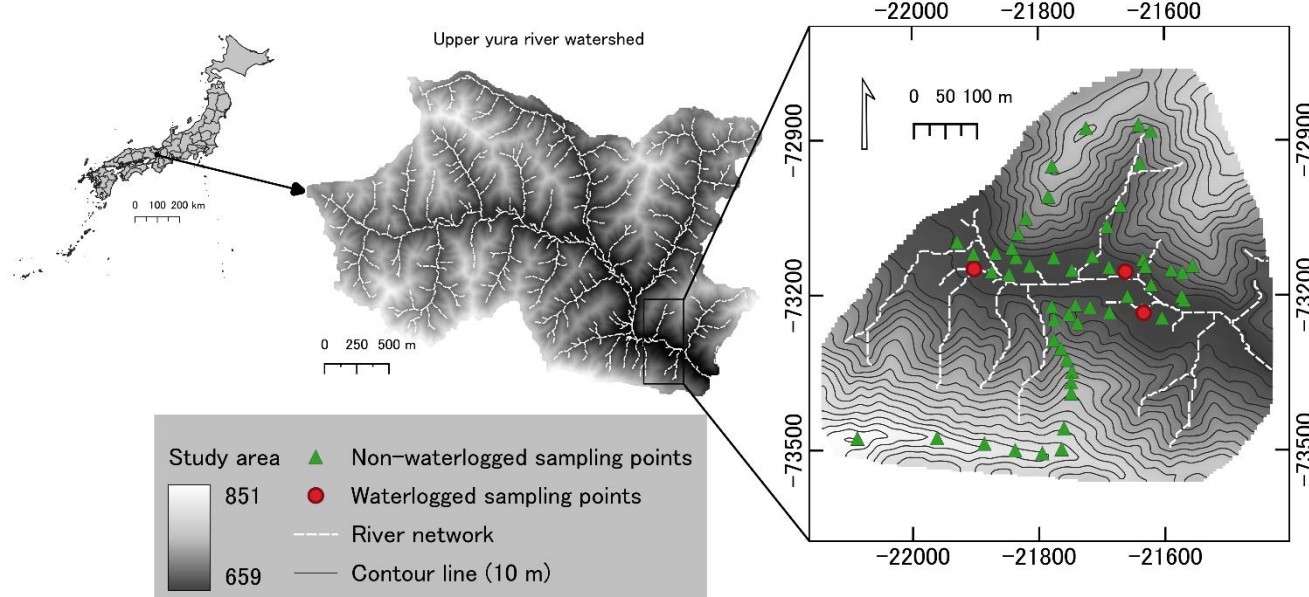

**Figure 1. Map of the upper Yura River watershed with its location in Japan on the left and an enlargement of the 40.2 ha study area on the right. The green triangles represent the 52 flux measurement locations on unsaturated soils and the red dots the 3 measurement locations on waterlogged soils. (Japan map: http://www.gsi.go.jp/ENGLISH/page_e30286.html)**

The study site covered an area of 40.2 hectares and included 55 sampling points for $CH_4$ flux measurements and soil sampling (Fig. 1). The sampling points were chosen along three transects perpendicular to the main river, from the plain to the ridges covering two slopes (south-facing and north-facing), as well as in a lateral canyon, and along transects parallel to the main river, on the plain, above the foot slope and on a ridge. The sampling was designed to encompass the landscape heterogeneity, while being constrained by the geography of the site and safety considerations. We recorded the positions of all sampling locations using a portable GPS tracker (Garmin, eTrex® Touch 35) accurate to a radius of 5 m or less.

## 2.2 Soil sampling and analysis

Soil cores were collected using a sampling cylinder at 0-10 cm depth at approximately 0.3 m of the flux measurement points. Samples were sieved at 2 mm and separated into stones and fine earth. The fresh weight of the fine earth fraction was measured before being air-dried. Bulk density of this fraction was determined as the ratio of oven-dried soil (subsample dried at 105°C) to the soil volume. Soil texture was analysed using the micro-pipette method, following Burt et al. (1993). Total soil carbon (C) and nitrogen (N) contents were measured using a Macro Corder JM 1000CN (J-SCIENCE LAB Co., Ltd., Japan). The soil pH was measured in a suspension (10 g of soil in 25 ml distilled $H_2O$) after shaking for 1 hour.

## 2.3 Topographic characterization

To characterize and process the terrain attributes related to soil $CH_4$ fluxes, we used a 0.5 m grid digital elevation model (DEM) based on airborne laser surveys conducted throughout the upper Yura River watershed in 2012 by the Ashiu Experimental Forest staff. The DEM was further processed and conditioned into a 5 m grid DEM image according to the GPS tracker's accuracy ($\leq 5$ m) that was used to locate each soil collar position, enabling us to identify the corresponding pixels on the terrain attribute grids. We derived several topographic attributes from the DEM using SAGA Next Generation in QGIS (v3.34.5-Prizren). The calculated attributes included slope, profile curvature (PrC), topographic position index (TPI), SAGA wetness index (SWI), and vertical distance to channel network (VDCN). Among the many attributes that can be derived from a DEM, we avoided selecting those that would be redundant to limit collinearities and overparameterization. Our preselection was motivated by the fact that methane fluxes result from the activity of methanotrophic and methanogenic communities, which are controlled by soil moisture and chemistry (C, N, pH), and, to a lesser extent, temperature. All the preselected attributed were correlated with soil moisture and chemistry (Table A1) and can potentially serve as a proxy for the spatial distribution of soil moisture and nutrient availability (Jeong et al., 2017; Kemppinen et al., 2018).

Slope, and profile curvature were calculated following the 9-parameter $2^{nd}$ order polynom method (Zevenbergen and Thorne, 1987). Negative values of profile curvature indicate a convex surface where the flow of water accelerates as it moves downslope; in contrast, positive values suggest a concave surface where the flow slows down (Pachepsky et al., 2001).

SWI is a refined version of the topographic wetness index (TWI) (Beven and Kirkby, 1979), which indicates that the spatial distribution of soil moisture is defined as $TWI = \ln(SCA/\tan\theta)$, where $SCA$ refers to the specific catchment area and $\theta$ is the local slope. SWI considers small differences in elevation values by using an iterative modification of the specific catchment area, assuming rather homogenous hydrologic conditions in the flat areas. The SWI was calculated using the SAGA wetness index algorithm, which is available in the SAGA library and integrated within QGIS (Conrad et al., 2015). Prior to computation, the DEM was hydrologically corrected by filling sinks to ensure continuous flow routing (Wang and Liu, 2006).

TPI describes the relative position of a location within a landscape, indicating whether it is on a ridge, slope, or valley based on the elevation compared to the surrounding terrain at a specified radius (Ågren et al., 2014). Positive values indicate ridges; negative values indicate depressions, and zero or near-zero values indicate slopes or flat areas. TPI is a highly scale-dependent variable and was calculated at the centre of circular areas with radii of 20 m, 30 m, and 50 m using the unfilled DEM. In our final model, we used TPI calculated with a 30 m radius, as it had the highest Spearman correlations with soil physical and chemical properties that influence soil $CH_4$ fluxes (Table A1).

VDCN was calculated as the elevation difference between each grid cell and the baseline of the nearest stream channel. This parameter serves as a proxy for groundwater depth, with lower VDCN values typically corresponding to areas with shallower groundwater and higher water tables, and higher values indicating deeper groundwater levels often found at higher topographic positions (Bock and Köthe, 2008). To calculate VDCN, the filled DEM was first used to create a flow accumulation layer using the multiple flow direction method (Freeman, 1991). The resulting flow accumulation raster was then used to create topographically defined flow channel networks, applying flow initiation thresholds of 0.5, 2.5, and 5 ha. VDCN then

subsequently calculated for each threshold. In our final model, we used VDCN calculated with a 5-ha initiation threshold, as it has the highest Spearman correlations with soil physical and chemical properties that influence soil $CH_4$ fluxes (Table A1). The site was classified into non-waterlogged areas (including ridges, slopes, foot slopes and plains as topographic positions where the soil is almost always unsaturated), wetlands (small patches with water-saturated soil year-round in the plain), and rivers. To distinguish wetland and non-waterlogged areas, we collected additional GPS positions at the edges and within the three wetland patches, in addition to the positions of the 55 sampling points. We then used SWI, PrC, slope, and VDCN to predict the locations of wetlands using a machine learning approach described in the supplementary file (Note S1). Finally, the boundaries between wetlands and non-waterlogged areas were refined by visual inspection. We acknowledged that using a fixed boundary between non-waterlogged areas and wetlands, although these boundaries may vary seasonally depending on the balance between precipitation and evaporation, may increase uncertainties in $CH_4$ flux prediction. Predicting the temporal variations of these boundaries was beyond the scope of this work, and, at our site, wetlands represent only 1% of the pixels (see below), and their boundaries even less. A posteriori, pixels classified as wetland had SWI values above 8.1, profile curvature between -0.003 and 0.001, slope values below 6.8, and VDCN values below 2.2 (Fig. S1). For river mapping, pixels corresponding to rivers were identified in the channel network raster, which was calculated using a 5-ha initiation threshold. Slope angle and TPI at 30 m radius were used to partition the non-waterlogged areas into ridges, slopes, foot slopes, and the plain. Locations with TPI values of 5 or greater were defined as ridges, representing locally elevated, convex surfaces. Locations with TPI values ≤ -5 were defined as foot slopes, concave surface. Areas with intermediate TPI values (−5 < TPI < 5) were further divided according to slope angle: sites with slope > 18° were defined as slopes, and those with slope ≤ 18° were defined as plains. Non-waterlogged areas, wetlands, and rivers, accounted for 94%, 1%, and 5% of the total study area, with respectively 52 sampling points located in non-waterlogged areas, including 14 in plains, 9 in foot slopes, 16 in slopes, and 13 in ridges, while 3 were situated in wetland areas.

## 2.4 Vegetation classification

Tree inventory was conducted during the flux measurement period to classify the vegetation surrounding the flux measurement points. A circular plot with a 10-meter radius was established, centred at each flux measurement point. Within the plot, all trees were identified at the species level, and their diameter at breast height (DBH) was measured. We calculated the plot basal area (BA) as the sum of the cross-sectional areas (CSA) at breast height of all tree trunks in each plot, and subsequently determined the relative basal area of coniferous trees ($RBA_{CON}$) in each plot. Then, we predicted the BA and $RBA_{CON}$ for the entire study area using SWI, TPI, VDCN, and the normalized vegetation index (NDVI) using a machine learning approach described in the supplementary file (Note S2, Fig. S2). Vegetation density was classified into three categories based on the quantile distribution of BA: high (BA > 2.6, upper quartile), medium (0.9 < BA < 2.6, interquartile range), and low (BA > 2.6, lower quartile). High, medium, and low vegetation density accounted for 37%, 37% and 26% of the total study area (Fig. S3), represented by 14, 28 and 10 sampling points, respectively. Vegetation types were classified based on $RBA_{CON}$. Three types were defined: coniferous when $RBA_{CON}$ was higher than 0.75, broadleaf when it was lower than 0.25, or mixed (comprising both conifers and broadleaved trees). These three types accounted 6%, 22% and72% of the total study area (Fig. S3), represented by 11, 19 and 22 sampling points, respectively.

## 2.5 Flux measurements

Soil $CH_4$ fluxes were measured using a static, non-steady-state, non-flow-through system composed of a dark acrylic chamber (20 cm diameter and 12.5 cm height) connected to a cavity-enhanced absorption spectroscopy gas analyser (Li 7810, Licor; Lincoln, USA) with two PTFE tubes, each 1.8 m long and 4 mm in inner diameter. One week before the first measurements, a 20 cm diameter, 9 cm tall PVC collar was inserted approximately 5 cm into the soil at each sampling point. Flux from each collar was measured on nine occasions in 2023: in early spring after snowmelt (4/27), mid-spring (5/12), late spring (5/31),

early summer (7/06), mid-summer (7/26), late summer (9/04), early autumn (10/07), mid-autumn (11/07), and late autumn (11/30). When measuring fluxes from the three small wetland patches, we took care to avoid trampling the soil near the collars, taking advantage of the abundant presence of stones and coarse woody debris.

To measure soil $CH_4$ flux, the chamber was placed on the collar, and changes in $CH_4$ and $CO_2$ concentrations inside were recorded for 4 minutes at a frequency of 1 Hz. The slope of the linear regression of $CH_4$ concentration over time was used to calculate the soil $CH_4$ flux:

$$F_{CH_4} = \frac{\Delta[CH_4]}{\Delta t} \times \frac{V \times P}{A \times R \times T}$$

where $F_{CH_4}$ is the soil $CH_4$ flux, $\frac{\Delta[CH_4]}{\Delta t}$ is the slope of the linear change in $CH_4$ concentrations over time, $V$ is the system volume (chamber, collar above the ground, tubes, and analyser), $A$ is the soil area covered by the collar, and $R$ is the ideal gas constant (8.314 J K$^{-1}$ mol$^{-1}$). A constant value of 93,525 Pa for an elevation of 650 m was used for the atmospheric pressure ($P$). The slope was calculated over 90 seconds following Epron et al. (2023). The $R^2$ of the linear variation of $CH_4$ concentration was less than 0.9 for a single measurement, and for this measurement, the $R^2$ of the linear variation of $CO_2$ concentration was 0.99, indicating that the low $R^2$ for $CH_4$ was due to the near-zero $CH_4$ flux and not to an erroneous measurement.

Soil moisture content and soil temperature near each collar were recorded on each measurement date using a soil moisture probe (SM150-T Device, Cambridge, UK) and a digital thermometer.

## 2.6 Climatic data

Air temperature and rainfall were measured every 10 minutes at a nearby weather station operated by the Field Science Education and Research Centre of Kyoto University. The antecedent precipitation index (API), an indicator of soil moisture conditions, was calculated using the following equation:

$$\text{API}_n = \sum_{t=1}^{n} P_t \times k^t$$

where, Pt is the precipitation during day t, k is the recession coefficient, and n is the number of antecedent days. The parameter k accounts for the water removed from the soil by evapotranspiration and drainage.

## 2.7 Modelling

We applied quantile regression forests (QRF) introduced by Meinshausen (2006), an extension of the random forests (RF) algorithm. RF is an ensemble learning method that builds a set of regression trees, and the final prediction is the average of all the regression trees, which are evaluated using out-of-bag cross-validation (Breiman, 2001). The QRF algorithm estimates the full conditional distribution of the response variable as a function of its predictors, not just the mean as with the original RF algorithm. Therefore, it is possible to extract the prediction interval for each pixel across the landscape for each measurement period. We used the five terrain attributes (slope, PrC, TPI at 30-m radius, SWI, and VDCN at 5-ha initiation threshold), basal area (BA), and relative basal area of coniferous trees to BA (RBA$_{CON}$) as predictors. Our strategy was to directly predict $CH_4$ fluxes using topographic and vegetation variables as proxies for soil moisture and chemistry, because incorporating soil moisture and chemistry as predictors, which would need to be extrapolated to the landscape level, would introduce additional layers of uncertainty. Unfortunately, the machine learning model was unable to accurately reproduce the measured fluxes when wetland measurements were included in the training dataset, likely due to the imbalance between the 52 non-waterlogged and only 3 wetland sampling points. The comparison of models including and not including wetland data is shown in Table A2 (3 of 55 collars for data, less than 1% of the landscape pixels). Patches, which had temporarily water-saturated soils, were not excluded.

We followed three steps to develop models for predicting soil $CH_4$ fluxes at each measurement period. Before applying QRFs, we eliminated the less important variables and identified the most relevant predictors for each measurement date, using a

variable selection algorithm for random forest models proposed by Genuer et al. (2010) and implemented in the "VSURF" package for R (Genuer et al., 2015). This approach systematically uses a repeated cross-validation procedure to rank variables by their importance index and iteratively eliminates the least informative ones to minimize model error. The result is a refined subset of predictors that enhances model interpretation and predictive performance. The predictor reduction approach has previously been used to map $CH_4$ fluxes (Räsänen et al., 2021; Warner et al., 2019) and soil properties (Jeong et al., 2017; Miller et al., 2015).

After selecting the relevant predictor variables, the QRF models were trained to predict $CH_4$ fluxes using the R-packages "caret" (Kuhn and Johnson, 2013) and "quantregForest" (Meinshausen, 2017). The mtry parameter, which determines the number of randomly selected predictor variables at each node, was tested from 2 to n-1 (n being the total number of predictors) using leave-one-out cross-validation to minimize prediction error and maximize the variance explained by the model. The ntree parameter was set to 500, ensuring the model constructed an ensemble of 500 decision trees. For each of the nine measurement dates, model accuracy was evaluated based on the root mean square error (RMSE) and coefficient of determination ($R^2$). $R^2$ was calculated as the square of the correlation between observed and cross-validated predicted fluxes, as implemented in the "caret" package. Furthermore, we calculated the variable's importance scores using the "vip" R-package (Greenwell and Boehmke, 2020). Variable importance scores were estimated using a permutation-based approach, in which the values of each predictor in the training data were randomly permuted to assess the resulting change in model performance, as quantified by the adjusted R-squared value. A greater reduction in adjusted $R^2$ indicated a higher importance of the predictor variable. We generated the accumulated local effect (ALE) plots to visualize the response of $CH_4$ fluxes to the predictor variables, accounting for the effect of the predictors in the model (Apley and Zhu, 2020). In ALE plots, an ALE value of zero on the y-axis corresponds to the mean predicted $CH_4$ flux, with positive values indicating higher and negative values indicating lower flux under the specific predictor on the x-axis. ALE reduces a complex machine learning function to depend on only one or, in some cases, two input variables, and visualizes the effect of a selected variable on the predicted $CH_4$ flux. The method removes the confounding effects of other input variables, computes the partial derivatives (local effects) of the prediction function with respect to the variable of interest, and integrates (accumulates) these effects across the range of that variable.

The output of the QRFs was a set of conditional prediction distributions of $CH_4$ fluxes for each landscape pixel and measurement dates. Because these prediction distributions were often not normally distributed, the median of the conditional prediction distribution at each pixel was used as the final prediction, and the interquartile range of the distribution was used to quantify the uncertainty in the prediction (Warner et al., 2019). Prediction uncertainties were expressed as a percentage (i.e., interquartile range of the conditional prediction distribution divided by the median). Modelling was conducted independently for each of the nine measurement dates, without including meteorological data, as in previous studies (Vainio et al., 2021; Warner et al., 2019).

**2.8 Statistical analysis**

We used analysis of variance (ANOVA) to test the differences in soil properties across the topographic positions and vegetation types and densities. Interactions were not included because the model would be rank-deficient as there were no "pure" broadleaved plots on the ridge. We examined the relationships between soil properties and topographic and vegetation variables using Spearman's rank correlation analysis using the 'Hmisc' package (Harrell Jr, 2003). Linear mixed-effect models (LMM) were used to test the relationship between the predicted fluxes at pixel levels and measured fluxes (fixed effect), with flux measurement dates as a random effect and between the predicted soil $CH_4$ fluxes and measured soil $CH_4$ fluxes aggregated by landscape units (topographical position, vegetation types, and vegetation density), which were included as random effects on both slope and intercept. The root mean square error (RMSE) was used to evaluate model performance at each date, and the marginal and conditional coefficients of the determination ($R_m^2$ and $R_c^2$) were used to determine the strength of the relationship between the predicted and measured fluxes. LMM was carried out using the 'lmerTest' package (Bates et al., 2015; Kuznetsova

et al., 2017), and $R_m^2$ and $R_c^2$ were calculated using the 'MUMIn' package (Bartoń, 2010). To test the effects of topographic positions, vegetation types, and densities on predicted $CH_4$ fluxes while accounting for spatial autocorrelation, we also used a linear mixed-effect model. Topographic positions, vegetation types, and densities were included in the model as fixed effects, and pixel ID as a random effect. Interactions were not included as no pixel contains "pure" broadleaved vegetation on the ridge. To eliminate spatial autocorrelation among residuals, we incorporated an exponential spatial correlation structure based on each pixel coordinate nested within each measurement date. This was performed using the 'nlme' package (Pinheiro et al., 1999). The semi-variogram of the residuals confirmed that the residuals were not spatially correlated. To quantify the effect size that indicates the relative contribution of each factor to the total variance in the response variable, we calculated eta-squared ($\eta_p^2$) values using the 'effectsize' package (Ben-Shachar et al., 2019). A pairwise comparison across the topographic positions, vegetation types, and densities was performed using the 'emmeans' package (Lenth, 2017). Linear regression models were used to examine the relationship between predicted soil $CH_4$ fluxes at the landscape scale and API. The recession coefficient (k) and the number of antecedent days (n) were not fixed a priori but optimized to maximize $R^2$ while ensuring the best distribution of the residuals, allowing parameters k and n to vary iteratively from 0.6 to 0.9 with an increment of 0.01 and from 0 to 30 with an increment of 0.01, respectively. Using a more complex bivariate model with an exponential function of air temperature did not improve the quality of the fit and returned $Q_{10}$ values that were not significantly different from 1, as previously reported (Epron et al., 2016). Calculations, modelling, and statistical analyses were performed using the R statistical programming environment (R Core Team, 2024).

## 3 Results

### 3.1 Environmental conditions and soil properties across non-waterlogged topographic and vegetation features

The total rainfall in the study area during the snow-free period of 2023 was 1578.5 mm, with relatively high rainfall in late-May to mid-June and a peak on August 15 due to the typhoon Lan (Fig. 2a). The monthly mean air temperature ranged from 7.5 to 24.2 °C during the study period (Fig. 2b). Mean soil moisture content varied seasonally, with the highest (47.7 ± 1.1 %; mean ± standard error) observed in the early summer (07/06) and the lowest (32.9 ± 1.2 %) in the late summer (09/04) (Fig. 2c). Mean soil temperature followed a similar trend to air temperature across the study period (Fig. 2d). Non-waterlogged soils consistently absorbed $CH_4$ (negative fluxes, Fig. 2e), while soils in the three small wetland patches emitted $CH_4$ (positive flux, Fig. A1). Variation in $CH_4$ fluxes across the measurement dates was consistent with the seasonal patterns of rainfall and air temperature. The fluxes measured on two collars that were temporarily waterlogged were positive on one occasion each.

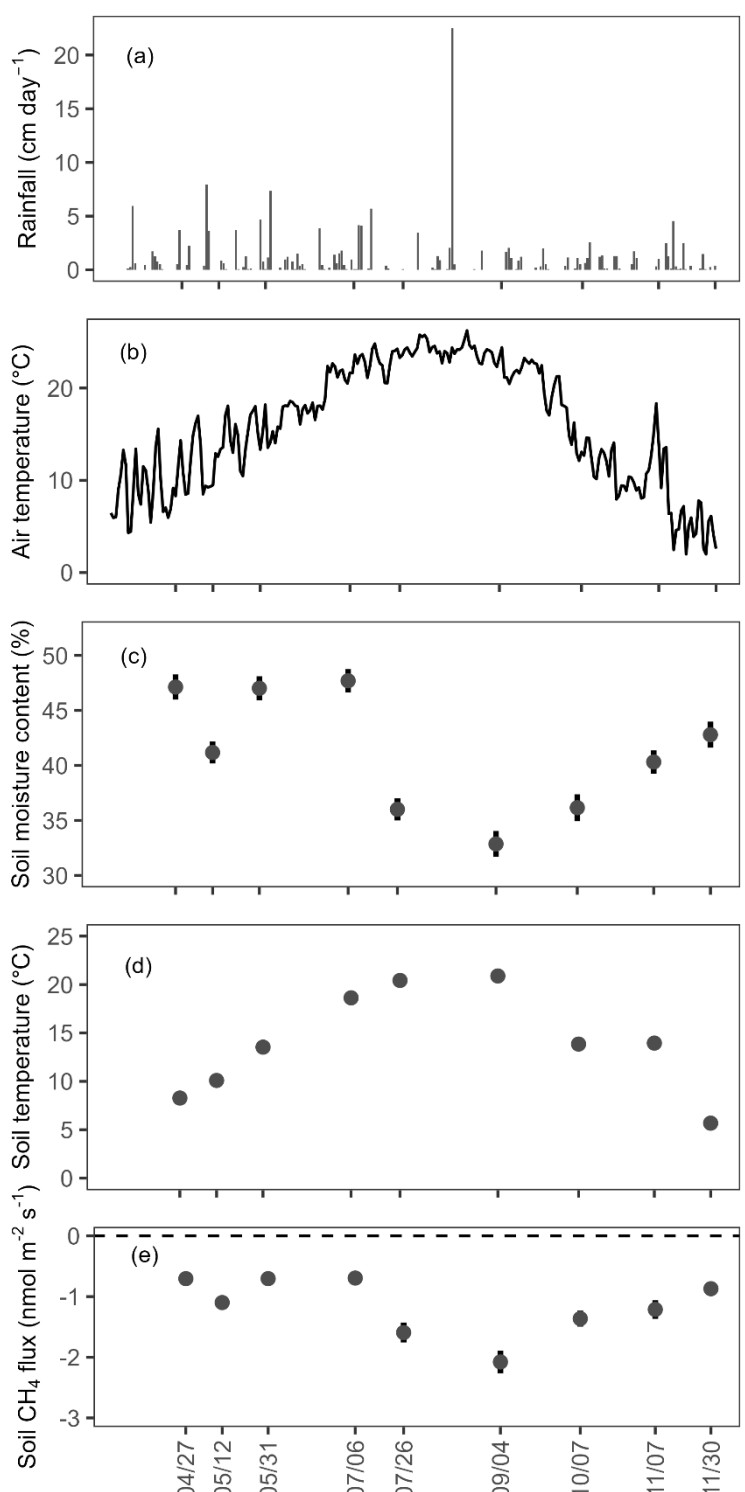

**Figure 2. Seasonal variation in (a) daily rainfall and (b) daily air temperature from April to November in 2023 measured at a weather station located nearby our study area, and (c) mean soil moisture content, (d) mean soil temperature, and (e) mean CH₄ fluxes from non-waterlogged soils, including all topographic positions (n = 52). Vertical bar indicating the standard error.**

310

Topographic positions were significantly related to several soil properties (bulk density, pH, total carbon and nitrogen, and mean temperature), whereas vegetation type and vegetation density were significantly related to soil temperature and soil moisture, respectively (Table 1). The bulk density of the fine earth fraction was relatively low due to the presence of stones, highest in the plain ($0.42 \pm 0.04$ g cm⁻³, mean ± SE) and significantly lowest in the ridge ($0.26 \pm 0.04$ g cm⁻³). Soil pH differed

significantly across topographic positions (p < 0.001), with more acidic conditions observed at higher elevations (ridge: $4.0 \pm 0.14$) compared to the plain ($5.1 \pm 0.73$). Similarly, total carbon (C) and total nitrogen contents (N) were significantly higher

on the ridges (16.7 ± 2.2% C and 0.8 ± 0.10% N) and lower in the plain (5.1 ± 0.73% C and 0.3 ± 0.04% N). Mean soil temperature also significantly varied across topographic positions, with the highest in the plain (14.3 ± 0.2°C) and the lowest in the foot slope (13.4 ± 0.2°C). In contrast, the soil texture of the fine earth fraction (clay, silt, and sand) and mean soil moisture content did not vary significantly with topographic positions.

Vegetation type significantly influenced soil temperature, with the highest values observed under broadleaved stands and the lowest under coniferous stands. Vegetation density significantly affected soil moisture content, which was highest in low-density areas and lowest in medium-density areas.

**Table 1. Mean (± standard error) of soil bulk density (BD), texture (clay, silt and sand), total carbon (C) and nitrogen (N) concentration, pH, mean soil water content (SWC) and temperature (T$_{soil}$) according to topographic position, vegetation type, and vegetation density. SWC and T$_{soil}$ are the average of the 9 measurement dates. A soil core (0-10 cm depth) was sampled at approximately 0.3 m of each soil collar. Different lowercase letters indicate significant differences among topographic positions, vegetation types, and vegetation densities (p < 0.05). The p-values of the ANOVA are shown in the last rows. The number of independent replicates in each factor level is indicated in the first column.**

| Factor | Bulk density (g cm$^{-3}$) | Clay (%) | Silt (%) | Sand (%) | pH | Total C (%) | Total N (%) | Mean SWC (%) | Mean T$_{soil}$ (°C) |
|---|---|---|---|---|---|---|---|---|---|
| Position | | | | | | | | | |
| Plain (n=14) | 0.42 ± 0.04 a | 8 ± 2 | 27 ±3 | 65 ± 4 | 4.9 ± 0.1 a | 5.1 ± 0.7 a | 0.3 ± 0.0 a | 44.2 ± 2.0 | 14.3 ± 0.2 a |
| Foot slope (n=9) | 0.36 ± 0.06 ab | 8 ± 2 | 33 ± 4 | 60 ± 5 | 4.3 ± 0.1 b | 8.3 ± 2.2 a | 0.5 ± 0.1 ab | 42.1 ± 2.0 | 13.4 ± 0.2 b |
| Slope (n=16) | 0.29 ± 0.03 ab | 10 ± 2 | 29 ± 3 | 60 ± 4 | 4.3 ± 0.1 b | 10.0 ± 1.1 a | 0.6 ± 0.1 ab | 39.2 ± 1.4 | 13.9 ± 0.1 a |
| Ridge (n=13) | 0.26 ± 0.04 b | 9 ± 2 | 23 ± 4 | 68 ± 4 | 4.0 ± 0.1 b | 16.7 ± 2.2 b | 0.8 ± 0.1 b | 40.1 ± 2.0 | 13.9 ± 0.1 ab |
| Vegetation type | | | | | | | | | |
| Broadleaved (n=19) | 0.38 ± 0.04 | 10 ± 2 | 29 ± 3 | 61 ± 4 | 4.7 ± 0.1 | 6.8 ± 1.0 | 0.4 ± 0.1 | 43.3 ± 1.6 | 14.2 ± 0.1 a |
| Coniferous (n=11) | 0.32 ± 0.04 | 7 ± 1 | 27 ± 4 | 66 ± 5 | 4.3 ± 0.1 | 11.1 ± 2.4 | 0.6 ± 0.1 | 40.8 ± 1.7 | 13.4 ± 0.2 b |
| Mixed (n=22) | 0.29 ± 0.03 | 9 ± 1 | 26 ± 2 | 64 ± 3 | 4.2 ± 0.1 | 12.4 ± 1.6 | 0.6 ± 0.1 | 39.7 ± 1.3 | 13.9 ± 0.1 a |
| Vegetation density | | | | | | | | | |
| High (n=14) | 0.32 ± 0.04 | 9 ± 2 | 28 ± 3 | 64 ± 4 | 4.4 ± 0.1 | 10.6 ± 2.02 | 0.57 ± 0.1 | 40.4 ± 0.1 a | 13.7 ± 0.2 |
| Medium (n=28) | 0.30 ± 0.03 | 10 ± 1 | 28 ± 3 | 62 ± 3 | 4.3 ± 0.1 | 11.3 ± 1.34 | 0.59 ± 0.1 | 39.5 ± 1.0 a | 13.9 ± 0.1 |
| Low (n=10) | 0.43 ± 0.05 | 8 ± 2 | 26 ± 3 | 66 ± 4 | 4.7 ± 0.2 | 6.1 ± 1.27 | 0.40 ± 0.1 | 47.6 ± 2.1 b | 14.2 ± 0.2 |
| Anova results | | | | | | | | | |
| Position | p < 0.05 | p = 0.55 | p = 0.33 | p = 0.49 | p < 0.001 | p < 0.001 | p < 0.01 | p = 0.12 | p < 0.001 |
| Vegetation type | p = 0.81 | p = 0.30 | p = 0.92 | p = 0.66 | p = 0.75 | p = 0.90 | p = 0.99 | p = 0.52 | p < 0.001 |
| Vegetation density | p = 0.58 | p = 0.45 | p = 0.58 | p = 0.43 | p = 0.99 | p = 0.93 | p = 0.98 | p < 0.05 | p = 0.91 |

**3.2 Selected variables and performance of the non-waterlogged soil CH₄ flux models**

The topographic position index (TPI) was consistently selected in all seasons, with high importance scores, ranging from 0.54 to 0.88, depending on the measurement dates (Table 2). The SAGA wetness index (SWI) was selected for most measurement dates, except for two, where the vertical distance to the channel network (VDCN) was selected instead. SWI importance scores were higher in summer than in the other seasons. VDCN and profile curvature (PrC) were occasionally selected along with TPI and TWI. VDCN showed moderate to low importance scores, contributing mostly in mid-spring (0.66) and early autumn (0.58). PrC, although less consistently selected, played a role in specific seasons, particularly in early spring and mid to late autumn. Accumulated local effect (ALE) plots showed the direction of the variables' effects on soil CH₄ fluxes for each measurement date (Fig. A2). For the two most influential predictors, low CH₄ uptake rates were associated with low TPI values, while they were associated with high SWI values. The vegetation density (BA) was selected only on two dates in 2023/04/27 and 2023/10/07, without improving the model accuracy, so we did not include it the final models (Appendix Table A3).

**Table 2. Selected variables for the quantile regression forest (QRF) models applied to non-waterlogged soil CH₄ fluxes at each measurement date, along with the $R^2$ and root mean square error (RMSE) values to evaluate the accuracy of the models. Importance scores of the selected variables are shown in parentheses, indicating their contribution to predicting soil CH₄ fluxes.**

| Measurement dates | Selected variables | $R^2$ | RMSE (nmol $m^{-2}$ $s^{-1}$) |
|---|---|---|---|
| 2023/04/27 | SWI (0.57), TPI (0.67), PrC (0.58) | 0.53 | 0.52 |
| 2023/05/12 | TPI (0.80), VDCN (0.66) | 0.31 | 0.82 |
| 2023/05/31 | SWI (0.55), TPI (0.57), VDCN (0.42) | 0.43 | 0.48 |
| 2023/07/06 | SWI (0.73), TPI (0.60) | 0.40 | 0.50 |
| 2023/07/26 | SWI (0.80), TPI (0.69) | 0.37 | 1.02 |
| 2023/09/04 | SWI (0.74), TPI (0.85) | 0.40 | 1.18 |
| 2023/10/07 | TPI (0.88), VDCN (0.58) | 0.67 | 0.81 |
| 2023/11/07 | SWI (0.32), TPI (0.84), VDCN (0.12), PrC (0.45) | 0.59 | 0.66 |
| 2023/11/30 | SWI (0.32), TPI (0.54), VDCN (0.21), PrC (0.28) | 0.51 | 0.56 |

Model accuracy showed seasonal variation, with the highest obtained in early autumn ($R^2$ = 0.67; RMSE = 0.81 nmol $m^{-2}$ $s^{-1}$) and the lowest in mid-spring ($R^2$ = 0.31; RMSE = 0.82 nmol $m^{-2}$ $s^{-1}$; Table 2). The relationship between measured and predicted fluxes for each measurement date showed that estimated fluxes were close to the observed fluxes (Fig. 3a-i).

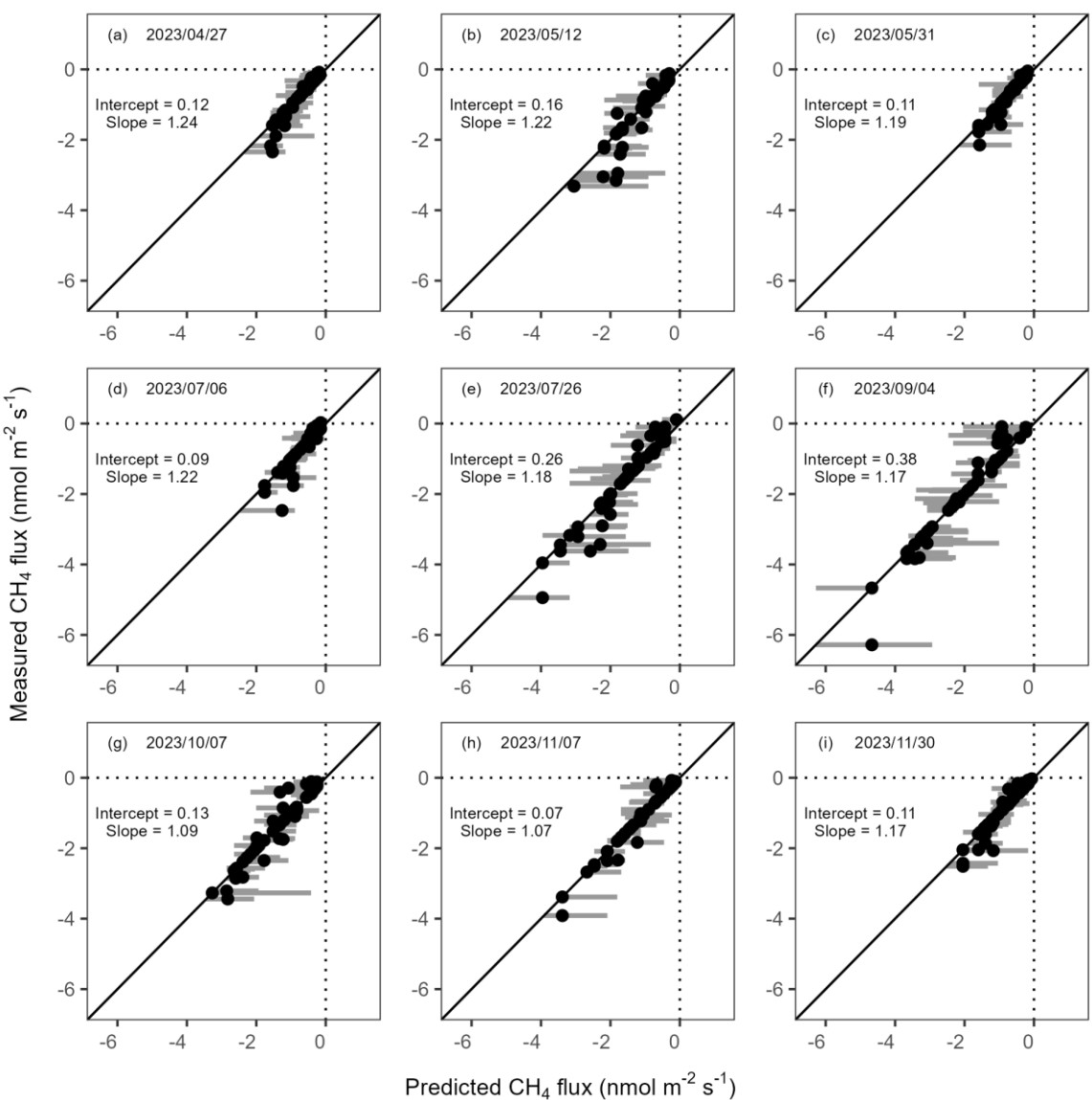

**Figure 3. Comparison of predicted (median of the quartile predictions from QRFs) and measured CH₄ fluxes (n = 52) for each measurement date. Vertical bars indicate the interquartile ranges of the prediction distribution. Intercepts and slopes are estimated using a linear mixed-effect model with measurement dates as a random effect (full statistics are shown in Table A4). The diagonals are the identity (1:1) lines.**

Overall, the slope of the relationship between measured and predicted fluxes (fixed effects) was not significantly different from 1 and was similar at all dates. The marginal ($R^2_m$) and conditional ($R^2_c$) coefficients of determination were 0.93 and 0.94, respectively, highlighting the consistency of the prediction for all measurement dates (linear mixed model, Table A4). To validate the fluxes at the landscape level, predicted fluxes were aggregated by landscape unit (i.e., topographic position, vegetation type, and vegetation density) and compared with the aggregated measured fluxes, which were consistent with the measured fluxes (Fig. 4, Table A5).

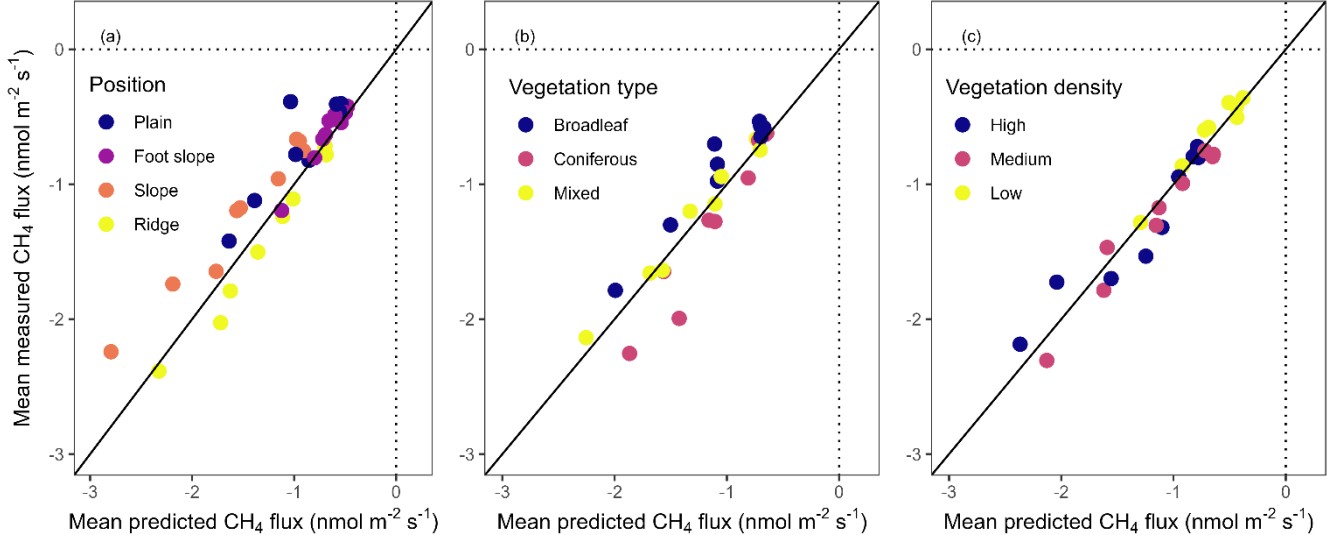

**Figure 4. Comparison between aggregated mean predicted and measured soil CH₄ fluxes from non-waterlogged soil for (a) topographic position, (b) vegetation type, and (c) vegetation density (full statistics are shown in Table A5). The diagonals are the identity (1:1) lines.**

### 3.3 Predicted non-waterlogged soil CH₄ fluxes

We predicted that non-waterlogged soils consistently uptake CH₄ across the seasons (negative fluxes, Fig.5). Predicted median CH₄ fluxes showed significant spatial heterogeneity, which was consistent across the seasons (Fig. 5). The highest net CH₄ uptake was predicted on ridges and the steepest parts of the slopes and decreased toward the foot slopes near streams and the flat plain (Fig. 6a). Coniferous and mixed stands showed the highest uptake compared to the broadleaved stands (Fig. 6b). Vegetation density (BA) also influenced the soil CH₄ uptake with higher uptake in the high and medium density areas. Although substantial variation was observed within each landscape unit, topographic position exerted the strongest control on CH₄ fluxes ($\eta_p^2 = 0.43$), followed by vegetation density ($\eta_p^2 = 0.11$) and vegetation type ($\eta_p^2 = 0.006$) (Table A6).

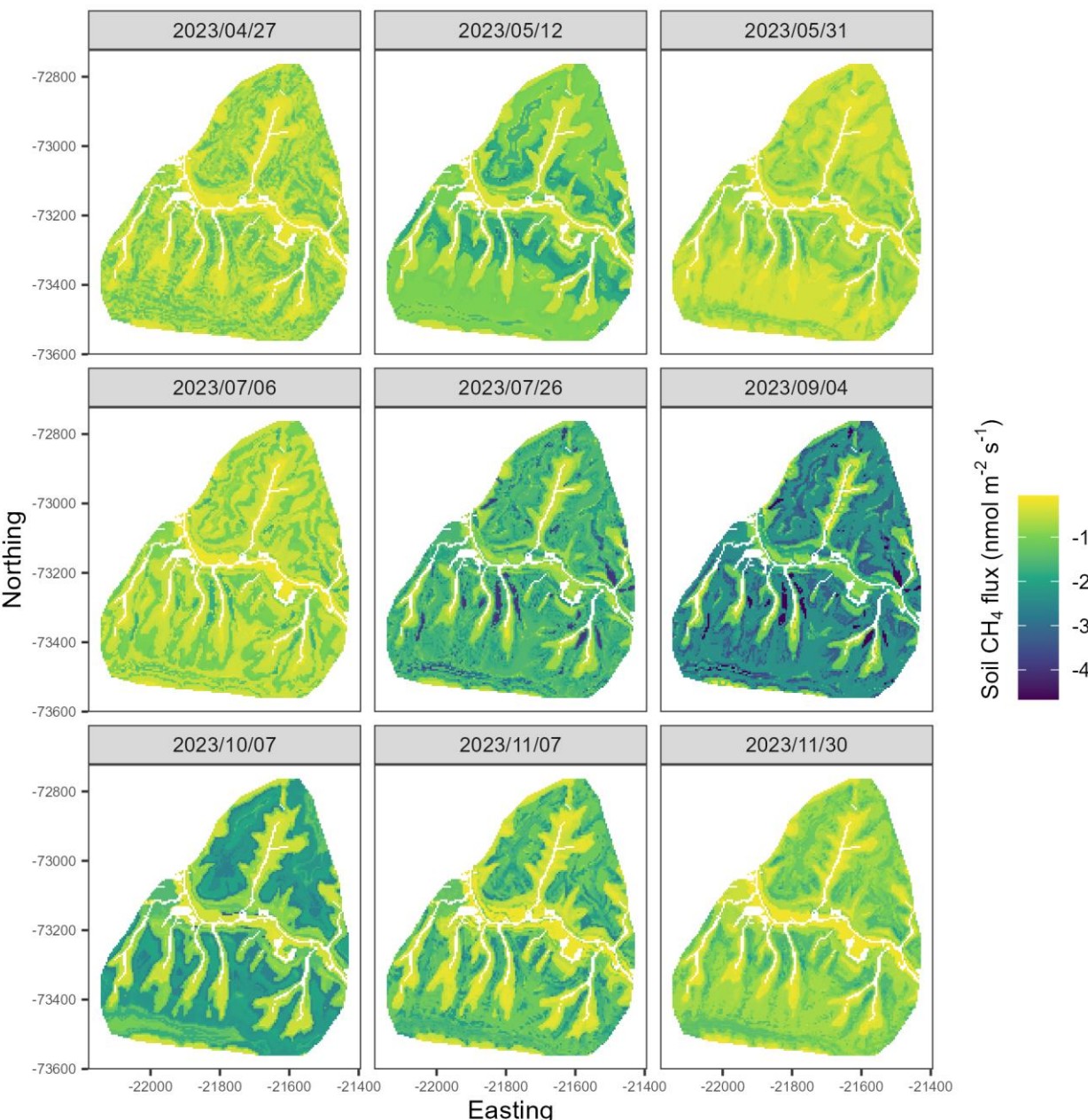

**Figure 5. Maps of predicted soil CH4 fluxes at each pixel of the study area (40.2 ha) for each measurement date. Values represent the median of the conditional prediction distribution for each pixel (5 m × 5 m).**

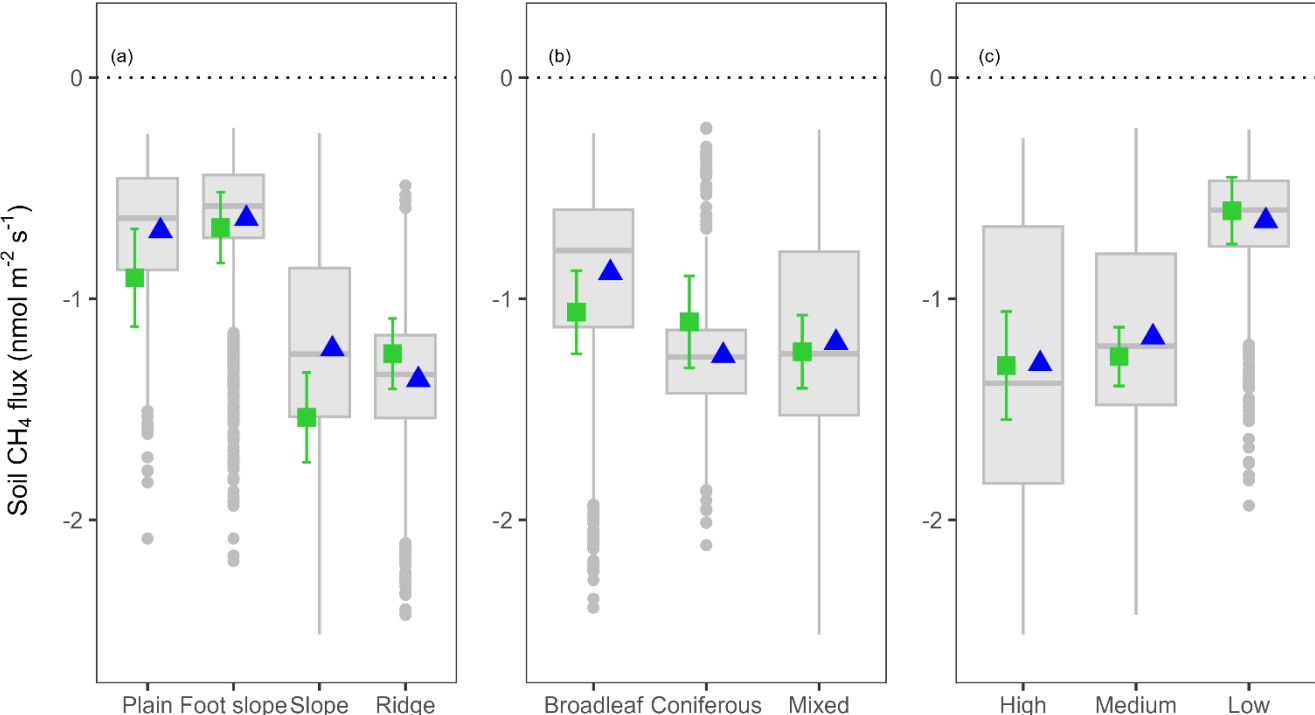

**385** **Figure 6. Predicted soil CH₄ fluxes at the landscape scale averaged over the nine measurement dates, aggregated by (a) topographic positions, (b) vegetation type, and (c) vegetation density (Full statistics with significance of the differences between each landscape unit shown in Table A6). Green squares indicate the mean of the measured fluxes with standard errors, blue triangles indicate the mean of the predicted fluxes at the landscape level and grey boxplots indicate the distribution of the predicted fluxes.**

**390** **3.4 Uncertainty of predicted non-waterlogged soil CH₄ fluxes**

The spatial distribution of the percentage of predicted uncertainty varied across seasons (Fig. 7). The percentage was consistently low to moderate (less than 100%) for pixels on ridges and steep slopes, but extremely high uncertainties (more than 500%) was observed at some dates for low-elevation pixels when predicted fluxes were close to zero. However, low predicted fluxes were often associated with equally low predicted uncertainty (Fig A3). The proportion of pixels with low

**395** uncertainty (<50%) was highest in early autumn (39.7% of the total non-waterlogged pixels) and lowest in early spring (5.7% of the total non-waterlogged pixels). In contrast, moderate uncertainty (50-100%) was predominant in most seasons, particularly in spring and autumn, accounting for approximately 50% of the landscape. Moderate to high uncertainty (101-500%) was also predominant on some measurement dates, particularly in late spring (49.8% of the total non-waterlogged pixels). Extreme uncertainty (>500%) was very rare in all seasons, except for a small peak in late autumn (0.26% of the total

**400** non-waterlogged pixels, Table A7).

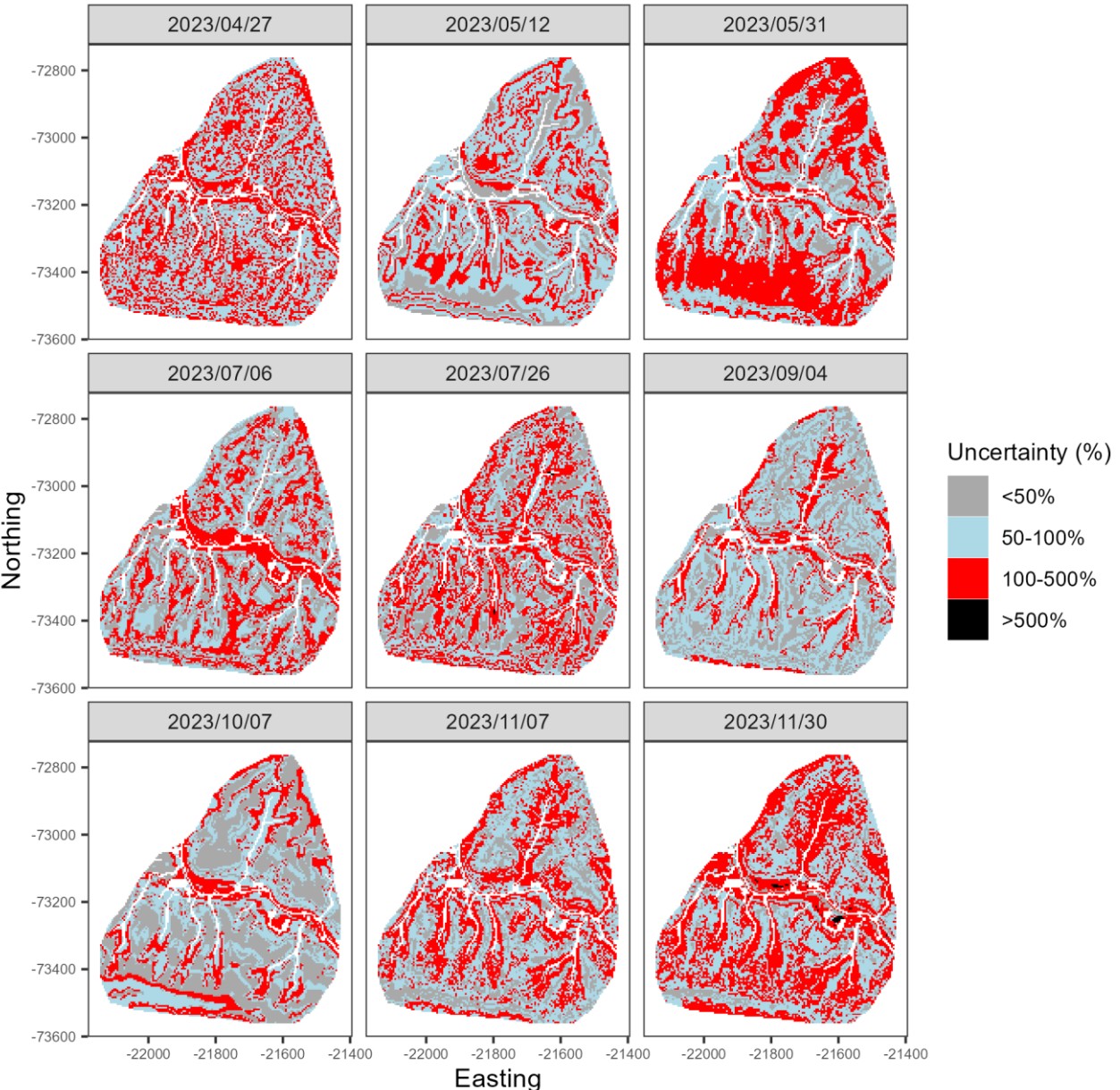

**Figure 7. Uncertainty map of predicted soil CH₄ fluxes at each pixel of the study area (40.2 ha) for each measurement date. Values represent the ratio of the interquartile range to the median of the prediction distribution for each pixel (5 m × 5 m).**

### 3.5 Predicted seasonal fluxes at the landscape level

The predicted CH₄ flux from non-waterlogged soil per hectare was calculated as the sum of the predicted fluxes at each pixel multiplied by pixel area (25 m²), and the sum divided by the non-waterlogged area. Across the landscape, the average CH₄ flux by non-waterlogged soils during the snow-free season was -0.66 (interquartile range: -0.94 to -0.44) g CH₄ ha⁻¹ hr⁻¹. Predicted median seasonal fluxes ranged from -0.34 to - 0.60 g CH₄ ha⁻¹ hr⁻¹ in spring, from -0.39 to -1.28 g CH₄ ha⁻¹ hr⁻¹ in summer, and from -0.48 to -0.89 g CH₄ ha⁻¹ hr⁻¹ in autumn (Fig. 8a). CH₄ uptake was low across the landscape in early (April 27) and late spring (May 31), while higher uptake was predicted in mid-spring (May 12). CH₄ uptake remained low in the early wet summer (July 6) and increased toward the mid (July 26) to late dry summer (Sep 4) when it reached its peaks. Net CH₄ uptake then decreased from early autumn (Oct 7) and reached its lowest rate in late autumn (Nov 30).

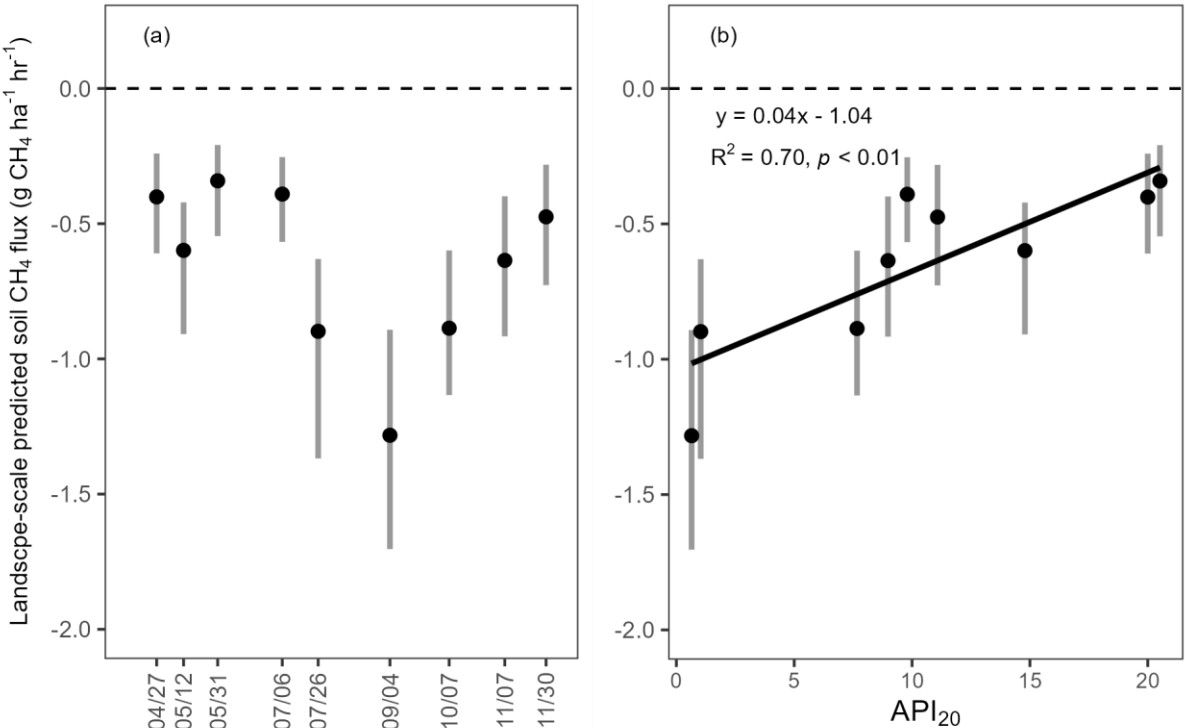

Figure 8. Predicted soil CH$_4$ fluxes, calculated as the mean of all pixels in the study area (40.2 ha), and antecedent precipitation index (API). (a) Seasonal variations in predicted soil CH$_4$ fluxes at the landscape scale and (b) relationship between predicted soil CH$_4$ fluxes at the landscape scale and the 20-day API. Vertical bars indicate the uncertainty of the predicted fluxes.

This seasonal variation in predicted median fluxes was well explained by the 20-day antecedent precipitation index ($R^2 = 0.70$, $p < 0.01$) with a recession coefficient of 0.69 (Fig. 8b), followed closely by the 30-day ($R^2 = 0.69$) and 7-day ($R^2 = 0.68$) API (Table A8).

## 4. Discussion

### 4.1 Selected variables

We employed quantile regression forest (QRF) models, driven only by topographic attributes, to upscale *in-situ* soil CH$_4$ flux measurements from sampling points to the landscape level for each measurement date in all topographic positions, but excluding wetlands (1% of our study area). Although selected twice, the inclusion of BA did not improve the accuracy and performance of the model and was eventually not retained in any final models, while RBA$_{CON}$ was never selected.

Among all tested topographic variables derived from the DEM, SWI, TPI, PrC, and VDCN were consistently selected in different models across all measurement periods, emphasizing their importance in upscaling CH$_4$ fluxes. Overall, the results validated our first hypothesis, as the selected topographic attributes were related to water circulation and accumulation.

Among these variables, SWI, which positively influence CH$_4$ fluxes (low uptake in areas with high SWI), represents water accumulation potential and is a common surrogate for soil moisture in mountainous regions. This key factor controls CH$_4$ fluxes by affecting gas diffusion and microbial activity (Kaiser et al., 2018; Vainio et al., 2021; Warner et al., 2019), as SWI integrates potential inflows and discharges through runoff and drainage (Ågren et al., 2014; Beven and Kirkby, 1979). SWI was selected in seven out of nine measurement periods but not on May 12 and October 7. These two periods correspond to transitional seasons, i.e., mid-spring and early autumn, when the landscape is generally drier, and water does not accumulate. TPI describes the elevation of a location relative to those of the surrounding terrain within a given radius, allowing the identification of landform positions such as ridges, slopes, and valleys (Ågren et al., 2014). TPI is generally calculated using a non-filled DEM, which is also more representative of local-scale moist depression that SWI doesn't capture, as SWI is

calculated using the filled DEM (Kemppinen et al., 2018). In our study, TPI was consistently selected in all measurement periods, and clearly related, highlighting that localized moisture, and potentially soil chemistry, are more influential parameters in controlling the $CH_4$ fluxes at the landscape level. Areas with negative TPI values (e.g., valleys or depressions) typically function as convergence zones, where water and nutrients accumulate due to gravitational flow and reduced drainage. In contrast, positive TPI values (e.g., ridges and convex upper slopes) are more divergent, often characterized by increased drainage and runoff, and limited water and nutrient retention. TPI negatively affected soil $CH_4$ fluxes (high uptake in areas with high TPI).

PrC refers to the curvature of the land surface in the direction of the slope (along a flow line) which was selected three times (April 27, Nov 7 and Nov 30) across the measurement dates. It influences the acceleration or deceleration of surface and subsurface water flow (Ågren et al., 2014). Negative values (concave slopes) tend to slow water movement, promoting water and nutrient accumulation in soils. Conversely, positive values (convex slopes) accelerate flow, often reducing water retention time and lowering nutrient accumulation due to leaching or erosion. Excluding PrC from the list of available variables for selection decreased the model performance for these three dates, probably because PrC helps discriminate between plains and slopes, both of which have near-zero TPI values.

VDCN is another important variable reflecting groundwater level conditions. Lower values typically observed near stream channels with higher groundwater level (Bock and Köthe, 2008). When the landscape was drier (May 12 and October 7), and SWI was not selected, TPI and VDCN had more substantial explanatory power. VDCN was also selected several times with SWI. Interestingly, VDCN has been shown to be useful in distinguishing well-drained from poorly drained soils (Bell et al., 1992; Kravchenko et al., 2002). It may explain why excluding VDCN from the list of variables available for selection decreased model performance. This highlights that SWI and TPI alone were not sufficient to reflect local soil moisture conditions, as drainage conditions can potentially vary across the landscape, which controls soil microhabitat conditions and thus influences $CH_4$ fluxes.

QRF modelling is non-parametric machine learning approach is particularly suited for handling non-linear relationships and complex interactions among predictors (Meinshausen, 2006). However, although the topographic predictors have successfully predicted $CH_4$ fluxes, the QRF method, like other statistical methods, does not provide a mechanistic understanding of the underlying biogeochemical processes, and the existence of confounding factors cannot be ruled out.

**4.2 Model performance and uncertainty**

Soil $CH_4$ fluxes predicted by QRF models were close to the measured fluxes for all measurement periods (Fig. 3; Table A4). We recognize that our models, by forcing pixel-scale predictors (5 m resolution) to explain finer-scale chamber measurements (20 cm diameter), may actually overestimate the predictive accuracy of the models at coarser scales. However, the predicted soil $CH_4$ fluxes not only closely matched the individual measured fluxes, but also when the two were aggregated by topographic position classes (ridge, slope, foot slope, and plain), vegetation density classes (high, medium, and low) or vegetation type classes (coniferous, mixed, and broadleaf, Fig. 4; Table A5). This confirmed that our models did not only predict point-level flux heterogeneity but were also able to capture the landscape-scale flux patterns and indicated that topographic attributes could be used for upscaling $CH_4$ fluxes in mountain landscapes. Overall, the performance of the models developed for scaling $CH_4$ fluxes was comparable to previous studies using topographic data for similar purposes (Kaiser et al., 2018; Vainio et al., 2021; Virkkala et al., 2024; Warner et al., 2019). However, it is important to note that direct comparisons between studies are difficult due to variations in cross-validation approaches, as the choice of cross-validation technique can significantly influence model performance (Roberts et al., 2017).

Unfortunately, it was not possible to accurately predict $CH_4$ fluxes when measurements collected in wetland patches were included in the training data, as the model accuracy decreased at all dates (Table A9). As a consequence, the marginal and conditional coefficients of determination of the relationship between the predicted and measured fluxes decreased from 0.93

and 0.94 respectively to 0.70 when wetland data were included. This is probably because neither the topographic features nor the vegetation differed sufficiently between the large areas functioning as $CH_4$ sinks and the small wetland patches in the plain area functioning as $CH_4$ sources (Fig. A1). Räsänen et al. (2021) noticed that spatial patterns of $CH_4$ fluxes could be accurately predicted in a northern peatland-forest-mosaic landscape when they were modelled for sinks and sources separately. This separation was not possible in our study due to the low number of measurement locations in wetlands, related to their small extent (1%) in our non-waterlogged soil-dominated landscape. Wetland exclusion, although acceptable in our 40-ha study area due to their small extent, would overestimate $CH_4$ uptake if incorrectly applied at larger scales, i.e., to the entire upper Yura River catchment in our case or to other hydrologically complex forest landscapes.

One advantage of the QRF approach is its ability to estimate prediction intervals (Meinshausen, 2006), thus offering insights into the uncertainty associated with the predicted flux value at each pixel. The spatial distribution of the uncertainty associated with the predicted soil $CH_4$ fluxes varied seasonally (Fig. 7; Table A7) in agreement with our second hypothesis, reflecting both spatial heterogeneity and temporal changes in model confidence. In our study, the spatial patterns of QRF-derived uncertainties were consistently related to topographic position and flux magnitude. Predictions in ridge and steep slope pixels generally exhibited low percentage uncertainties (often below 100%), likely because these well-drained areas were well represented in the training data and exhibited relatively stable and high $CH_4$ uptake across seasons. In contrast, extremely high percentage uncertainties (exceeding 500%) were observed in some low-lying pixels during specific seasons, especially where predicted $CH_4$ fluxes were close to zero. Our models did not predict median positive fluxes although positive fluxes were occasionally measured. However, the possibility of positive fluxes is reflected in the large uncertainties associated with near-zero fluxes. A crucial methodological point is that percentage uncertainty is a relative measure; even a small absolute uncertainty around a near-zero prediction can yield a very large percentage (Warner et al., 2019). In addition, large absolute uncertainties can result from large differences in fluxes measured at locations with similar topographic characteristics.

The lowest uncertainty was obtained in late summer and early autumn, i.e., under warm and dry conditions, indicating better model performance when hydrological conditions were less variable. In contrast, larger uncertainties were produced by the models in early spring and late autumn, as well as in late spring and early summer, when measured and predicted soil $CH_4$ fluxes were lowest. The East Asian monsoon flow bringing warm and humid air mass and resulting in the rainy season in late spring and early summer, as well as low evapotranspiration in early spring and late autumn, may have introduced greater variability in soil hydrology, contributing to higher uncertainties. Nevertheless, low to moderate uncertainty (<100%) was the most prevalent class across all seasons, consistently accounting for more than half the landscape—up to 80% in late summer and early autumn—while extreme uncertainties (>500%) were very rare in all seasons. This suggests that the models performed well overall. Although some areas remain challenging to model, the QRF approach provides generally reliable spatial predictions of soil $CH_4$ fluxes with quantifiable and interpretable uncertainties.

### 4.3 Spatial patterns of predicted soil $CH_4$ fluxes

The models revealed clear spatial patterns in soil $CH_4$ fluxes that were consistent across measurement dates, even though the models selected different variables at each date. Predicted soil $CH_4$ fluxes closely matched topographic gradients, consistent with our third hypothesis. Ridges and upper slopes exhibited the highest net $CH_4$ uptake, functioning as strong sinks for $CH_4$ across all seasons, whereas $CH_4$ uptakes were lowest in plain and foot slope positions. These topographic patterns of $CH_4$ uptake are consistent with previous studies. In a temperate forest in central Ontario, Canada, the highest $CH_4$ uptake was observed on slopes and ridges (Wang et al., 2013). Similarly, in a temperate forest in Maryland, USA, transition zones were identified as hotspots for $CH_4$ uptake (Warner et al., 2018). In a tropical forest in China, hillslopes exhibited the highest $CH_4$ uptake, while lower uptake was observed at the foot slopes and in groundwater discharge areas (Yu et al., 2021). Similarly, $CH_4$ uptake was greater on ridges than at valley bottoms in a subtropical forest in Puerto Rico (Quebbeman et al., 2022).

In our studied landscape, we observed lower soil bulk density on ridges and slopes than on the plain area, indicating that ridge and slope soils have higher porosity, which is consistent with higher soil $CH_4$ oxidation rates due to higher diffusion rates of $O_2$ and $CH_4$ from the atmosphere through soil pores (Ishizuka et al., 2009). Although we did not assess the methanotroph community structure, the greater atmospheric $CH_4$ uptake on slopes and ridges is consistent with the community structure observed in a subalpine forest, with type I methanotrophs dominating in riparian soils, whereas type II methanotrophs were more prevalent in upland soils (Du et al., 2015). The higher soil carbon (C) and nitrogen (N) contents observed on ridges and slopes at our site may contribute to higher soil $CH_4$ uptake, as soil $CH_4$ uptake has been found to be positively correlated with soil organic matter content in subtropical and temperate forests (Lee et al., 2023). Possible explanations are that higher soil carbon may increase the availability of labile substrates that stimulate methanotrophic activity by increasing $CH_4$ supply through enhanced methanogenesis in anoxic microsites or by directly providing substrate for facultative methane-oxidizing bacteria, thereby increasing their abundance (Jensen et al., 1998; Semrau et al., 2011; West and Schmidt, 1999). Soil nitrogen was probably predominantly in organic form, and therefore the soil concentration of nitrate and ammonium, known to inhibit $CH_4$ oxidation by methanotrophs at high concentration (King and Schnell, 1994; Mochizuki et al., 2012), likely remained low (Aronson and Helliker, 2010; Bodelier and Laanbroek, 2004). Nitrogen is an essential nutrient for the growth of methanotrophs, whose activity has been shown to be nitrogen-limited in forest soils (Börjesson and Nohrstedt, 2000; Martinson et al., 2021; Veldkamp et al., 2013). Therefore, mineralization of these low levels of organic nitrogen could alleviate the nitrogen limitation of $CH_4$ oxidation and partly explain the higher soil $CH_4$ uptake observed on ridges and slopes, where total nitrogen concentration was higher than at the foot slopes and in the plain.

Although the effect-size of vegetation density was much smaller than that of topographic position, the predicted soil $CH_4$ uptake was significantly lower in areas with low basal area. Vegetation density can also potentially be related to local moisture conditions, as dense vegetation likely consume more water, thus increasing the soil air-filled porosity (Hakamada et al., 2020; Vanclay, 2009). Unexpectedly, although a very small effect-size, our models predicted higher soil $CH_4$ uptake in conifer-dominated areas and lower uptake in broadleaf-dominated areas, contrary to previous evidence of greater soil $CH_4$ uptake in plots containing only deciduous broadleaved tree species than in plots containing evergreen coniferous trees, either alone or in mixture (Jevon et al., 2023). The discrepancy between this previous study and our results may be related to the fact that their study area was ten times smaller and more topographically homogeneous than ours (4 *versus* 40 ha). Moreover, soil properties that could explain the lower rate of $CH_4$ oxidation in coniferous than in broadleaved stands, such as higher acidity (Borken et al., 2003; Hütsch, 1998; Ishizuka et al., 2000) did not differ significantly among the three types of vegetation cover at our site, whereas they differed according to topographic position. However, vegetation types and density were not randomly distributed among topographic positions (Table A9), meaning that the confounding effects of vegetation and DEM-derived variables on the prediction soil $CH_4$ uptake could make it difficult to separate the influence of vegetation and topography in our complex mountain landscape.

## 4.4 Predicted soil $CH_4$ fluxes at the landscape scale and seasonal variation

The $CH_4$ fluxes per hectare were calculated by aggregating pixel-level predictions and normalizing them to the total non-waterlogged area, allowing for standardized comparison across sites, although there are still very few comparable data available, making it difficult to analyse the causes of differences across sites. Our highest $CH_4$ uptake in late summer was -1.28 g $CH_4$ ha$^{-1}$ hr$^{-1}$ (interquartile range -1.70 to -0.89), 2.6 times higher in absolute value than in a forested watershed in Maryland, USA (-0.47 g $CH_4$ ha$^{-1}$ hr$^{-1}$, Warner et al. 2019), but slightly lower than in a boreal pine forest in Finland (-1.59 g $CH_4$ ha$^{-1}$ hr$^{-1}$, Vainio et al. 2021).

Consistent with our fourth hypothesis, the seasonal variation in soil $CH_4$ fluxes at the landscape scale in the non-waterlogged areas demonstrates a strong sensitivity to soil moisture dynamics, which were effectively captured using the Antecedent Precipitation Index (API). The API, serving as a proxy for soil moisture dynamics, integrates precipitation over a defined

period and includes a recession factor to account for evapotranspiration and drainage. Short durations (e.g., 7 days) reflect surface moisture, while longer durations (e.g., 30 days) capture deeper soil moisture conditions (Schoener and Stone, 2020; Sidle et al., 2000; Yamao et al., 2016). Among the API durations tested, the 20-day API with a recession coefficient of 0.69 showed the highest explanatory power ($R^2$ = 0.70), although using either a 30-day or a 7-day API would provide similar goodness of fit with similar recession coefficients, indicating that soil moisture conditions across different depths had similar influence on $CH_4$ flux variability. The consistently low recession coefficient (Kohler and Linsley, 1951) suggested that rainwater does not accumulate in our watershed. High API values indicate wetter antecedent conditions, which can suppress $CH_4$ uptake by reducing oxygen availability and thus limiting methanotrophic activity, and by temporarily turning the subsoil condition to anoxic, promoting methane production and reducing net $CH_4$ uptake (Angel et al., 2012; Hu et al., 2023; Kruse et al., 1996). Conversely, drier periods with low API values were observed in mid and late summer and earlier autumn, when soils were better aerated, creating favourable conditions for atmospheric $CH_4$ oxidation and leading to greater $CH_4$ uptake.

**5 Conclusion**

In conclusion, our study showed the dominant role of topography, compared to that of vegetation, on the spatial variation of soil $CH_4$ fluxes in mountain forest landscapes throughout the snow-free season. The quantile regression forest models successfully captured these ridge-to-plain spatial gradients where the soil is almost always unsaturated, with strong performance. However, our modelling approach was unable to accurately predict $CH_4$ fluxes when including measurements collected in three wetland patches functioning as $CH_4$ sources in the plain area (1% of the total landscape). $CH_4$ uptake was consistently highest on ridges and slopes, where well-drained soils with lower bulk density and higher porosity supported enhanced methanotrophic activity. Furthermore, the seasonal dynamics of the predicted soil $CH_4$ flux at the landscape scale was well-captured by the 20-day Antecedent Precipitation Index (API), with a significant positive relationship between API and $CH_4$ uptake, emphasizing the sensitivity of $CH_4$ uptake by non-waterlogged soils to seasonal fluctuations in soil moisture conditions. The integration of terrain-based predictors and moisture history provides a reliable framework for scaling soil $CH_4$ fluxes across complex landscapes, highlighting the importance of considering both static (topography, vegetation) and dynamic (climate) controls in future assessments of $CH_4$ flux.

 **Appendix A**

**Table A1. Spearman's rank correlation test between soil properties measured on soil cores (0-10 cm depth) sampled at approximately 0.3 m of each soil collar and topographic and vegetation attributes. Significant coefficients are shown in bold and italic.**

|  | Mean SWC | Mean $T_{soil}$ | Total C | Total N | pH | BD | Sand | Silt | Clay |
|---|---|---|---|---|---|---|---|---|---|
| PrC | -0.21 | 0.14 | ***0.27*** | 0.23 | -0.27 | -0.14 | 0.24 | -0.26 | -0.06 |
| Slope | -0.17 | -0.24 | 0.25 | 0.22 | ***-0.51*** | -0.11 | 0.00 | -0.02 | 0.01 |
| TPI 20m | ***-0.35*** | -0.11 | ***0.45*** | ***0.39*** | ***-0.34*** | -0.26 | 0.05 | -0.04 | 0.03 |
| TPI 30m | ***-0.31*** | -0.08 | ***0.64*** | ***0.55*** | ***-0.46*** | ***-0.36*** | 0.11 | -0.14 | 0.03 |
| TPI 50m | ***-0.33*** | -0.08 | ***0.59*** | ***0.50*** | ***-0.42*** | ***-0.31*** | 0.11 | -0.13 | 0.02 |
| SWI | ***0.43*** | 0.27 | ***-0.62*** | ***-0.55*** | ***0.65*** | ***0.41*** | -0.03 | 0.04 | -0.01 |
| VDCN 0.5ha | -0.24 | -0.16 | ***0.61*** | ***0.52*** | ***-0.51*** | ***-0.36*** | 0.16 | -0.20 | 0.04 |
| VDCN 2.5ha | -0.24 | -0.20 | ***0.71*** | ***0.62*** | ***-0.65*** | ***-0.42*** | 0.18 | -0.22 | -0.02 |
| VDCN 5ha | ***-0.31*** | ***-0.29*** | ***0.65*** | ***0.56*** | ***-0.71*** | ***-0.47*** | 0.16 | -0.23 | 0.09 |
| BA | ***-0.37*** | ***-0.29*** | 0.22 | 0.13 | ***-0.25*** | -0.19 | -0.03 | 0.08 | 0.00 |
| RBA$_{CON}$ | -0.17 | ***-0.56*** | ***0.30*** | 0.19 | ***-0.41*** | -0.17 | 0.03 | 0.02 | -0.11 |

 **SWC: soil water content; $T_{soil}$: soil temperature; BD: soil bulk density**

**Table A2. $R^2$ and root mean square error (RMSE) values for the quantile regression forest (QRF) models applied to soil CH$_4$ fluxes without wetland and with wetland at each measurement date. Note that the same variables were selected at all dates in both cases.**

| Measurement dates | Selected variables | $R^2$ (RMSE, nmol m$^{-2}$ s$^{-1}$) | $R^2$ (RMSE, nmol m$^{-2}$ s$^{-1}$) |
|---|---|---|---|
|  |  | Without wetland | With wetland |
| 2023/04/27 | SWI, TPI, PrC | 0.53 (0.52) | 0.37 (0.86) |
| 2023/05/12 | TPI, VDCN | 0.31 (0.82) | 0.22 (1.27) |
| 2023/05/31 | SWI, TPI, VDCN | 0.43 (0.48) | 0.25 (1.07) |
| 2023/07/06 | SWI, TPI | 0.40 (0.50) | 0.34 (1.99) |
| 2023/07/26 | SWI, TPI | 0.37 (1.02) | 0.30 (1.60) |
| 2023/09/04 | SWI, TPI | 0.40 (1.18) | 0.38 (1.27) |
| 2023/10/07 | TPI, VDCN | 0.67 (0.81) | 0.42 (1.07) |
| 2023/11/07 | SWI, TPI, VDCN, PrC | 0.59 (0.66) | 0.47 (1.25) |
| 2023/11/30 | SWI, TPI, VDCN, PrC | 0.51 (0.56) | 0.40 (0.57) |

**Table A3. Comparison of the accuracy of the quantile regression forest (QRF) models applied to non-waterlogged soil CH₄ fluxes without and with vegetation at the two dates where BA was selected. Selected variables and their importance scores in parentheses, along with the $R^2$ and root mean square error (RMSE) values (model with vegetation is in bold italic letters).**

| Date | Selected variables | | | $R^2$ | RMSE |
|------|------|------|------|------|------|
| 2023/04/27 | SWI (0.57) | PrC (0.58) | TPI (0.67) | 0.53 | 0.52 |
| | *__SWI (0.64)__* | *__BA (0.42)__* | *__TPI (0.50)__* | *__0.51__* | *__0.52__* |
| 2023/10/07 | TPI (0.88) | VDCN (0.58) | | 0.67 | 0.81 |
| | *__TPI (0.77)__* | *__VDCN (0.39)__* | *__BA (0.27)__* | *__0.55__* | *__0.80__* |

**Table A4. Summary of the linear mixed model (LMMs) analysing the relationship between the predicted soil CH₄ fluxes and measured soil CH₄ fluxes, with measurement periods included as a random effect on both slope and intercept. The p-values of the fixed effect were for testing if the intercept was different from zero and the slope different from 1. The statistics panel at the bottom left shows the marginal ($R^2_m$) and conditional ($R^2_c$) coefficients of determination, the root mean square error of the model, and the overall significance of the model (p-value).**

| Fixed effect: predicted CH₄ flux | | | Random effects: measurement dates | | |
|------|------|------|------|------|------|
| Estimate ± SE | | p-values | | Intercept | Slope |
| Intercept | 0.15 ± 0.03 | from 0: 0.003 | 2023/04/27 | -0.05 | 0.03 |
| Slope | 1.16 ± 0.02 | from 1: 0.92 | 2023/05/12 | -0.02 | 0.02 |
| | | | 2023/05/31 | -0.05 | 0.02 |
| | | | 2023/07/06 | -0.06 | 0.02 |
| Statistics | | | 2023/07/26 | 0.07 | 0.00 |
| n | 467 | | 2023/09/04 | 0.14 | -0.03 |
| $R^2_m$ | 0.94 | | 2023/10/07 | 0.02 | -0.04 |
| $R^2_c$ | 0.95 | | 2023/11/07 | -0.01 | -0.04 |
| RMSE | 0.26 | | 2023/11/30 | -0.03 | 0.01 |
| p-value | $2.5 \times 10^{-9}$ | | | | |

**Table A5. Summary of the linear mixed model (LMMs) analysing the relationship between the predicted soil CH$_4$ fluxes and measured soil CH$_4$ fluxes, with landscape units (either positions, vegetation types, and vegetation density) included as a random effect on both slope and intercept. The p-values of the fixed effect were for testing if the intercept was different from zero and the slope different from 1. The statistics panel at the bottom left shows the marginal (R$^2$$_m$) and conditional (R$^2$$_c$) coefficients of determination, the root mean square error of the model, and the overall significance of the model.**

| Fixed effect: Predicted CH$_4$ flux | | | Random effect: Positions | | |
|---|---|---|---|---|---|
| Estimate ± SE | | p values | | Intercept | Slope |
| Intercept | -0.17 ± 0.04 | from 0: 0.006 | Plain | 0.01 | -0.02 |
| Slope | 0.94 ± 0.05 | from 1: 0.99 | Foot slope | 0.01 | 0.01 |
| Statistics | | | Slope | -0.05 | 0.1 |
| n | 36 | | Ridge | 0.05 | -0.09 |
| R$^2$m | 0.93 | | | | |
| R$^2$c | 0.98 | | | | |
| RMSE | 0.20 | | | | |
| p-value | $7.8 \times 10^{-05}$ | | | | |
| Fixed effect: Predicted CH$_4$ flux | | | Random effect: Vegetation type | | |
| Intercept | -0.19 ± 0.07 | from 0: 0.08 | Broadleaf | 0.02 | 0.02 |
| Slope | 0.82 ± 0.09 | from 1: 0.99 | Coniferous | -0.09 | -0.14 |
| Statistics | | | Mixed | 0.08 | 0.12 |
| n | 27 | | | | |
| R$^2$m | 0.91 | | | | |
| R$^2$c | 0.96 | | | | |
| RMSE | 0.17 | | | | |
| p-value | 0.01 | | | | |
| Fixed effect: Predicted CH$_4$ flux | | | Random effect: Vegetation density | | |
| Intercept | -0.13 ± 0.05 | from 0: 0.08 | High | -0.01 | 0.02 |
| Slope | 0.81 ± 0.08 | from 1: 0.99 | Medium | 0.06 | -0.14 |
| Statistics | | | Low | -0.05 | 0.11 |
| n | 27 | | | | |
| R$^2$m | 0.80 | | | | |
| R$^2$c | 0.95 | | | | |
| RMSE | 0.19 | | | | |
| p-value | 0.01 | | | | |

**Table A6. Summary of the linear mixed model (LMM) analysing the effects of topographic position, vegetation type, and vegetation density on predicted soil CH$_4$ fluxes. Pixel ID was included as a random effect, and spatial autocorrelation among residuals was eliminated. $\eta_p^2$ was calculated as the effect size of each explanatory variable. Letters indicated the significance within each landscape unit.**

| Explanatory variables | p-value | Effect size ($\eta_p^2$) [with 95% CI] |
|---|---|---|
| Position [df=3] | <0.001 | 0.43 [0.42, 0.44] |
| *Plain (a), Foot slope (b), Slope (c), Ridge (c)* | | |
| Vegetation type [df=2] | <0.001 | 0.006 [0.00, 0.01] |
| *Broadleaf (a), Coniferous (b), Mixed (c)* | | |
| Vegetation density [df=2] | <0.001 | 0.11 [0.10, 0.12] |
| *High (c), Medium (b), Low (a)* | | |

**Table A7. Percentage of pixels in the study area distributed among four levels of predicted relative uncertainty for soil CH$_4$ fluxes from non-waterlogged soil.**

| Measurement date | Uncertainty | | | |
|---|---|---|---|---|
| | < 50 % | 50 - 99 % | 100- 500 % | > 500 % |
| 2023/04/27 | 5.72% | 53.29% | 40.99% | 0.01% |
| 2023/05/12 | 19.93% | 54.01% | 26.06% | - |
| 2023/05/31 | 10.90% | 39.29% | 49.81% | - |
| 2023/07/06 | 27.76% | 38.32% | 33.92% | - |
| 2023/07/26 | 21.12% | 40.66% | 38.13% | 0.09% |
| 2023/09/04 | 30.19% | 51.24% | 18.58% | - |
| 2023/10/07 | 39.68% | 38.13% | 22.19% | - |
| 2023/11/07 | 17.50% | 46.10% | 36.4% | - |
| 2023/11/30 | 7.83% | 43.65% | 48.26% | 0.26% |

**Table A8. Statistics of the linear relationship between soil CH$_4$ fluxes at the landscape scale and antecedent precipitation indexes (API). 20 antecedent days provided the best fit. 30 and 7 antecedent days are shown as common metrics in hydrology. Adjusted recession coefficients (k) and determination coefficients (R$^2$) are shown.**

| Antecedent days | k | R$^2$ |
|---|---|---|
| 20 | 0.69 | 0.70 |
| 30 | 0.69 | 0.69 |
| 7 | 0.67 | 0.68 |

**Table A9. Proportion of vegetation density and type associated with the different topographic positions across the study area (40.2 ha).**

| Position | Vegetation density | Proportion (%) | Vegetation type | Proportion (%) |
|---|---|---|---|---|
| Plain | High | 19.6 | Broadleaf | 91.8 |
|  | Medium | 8.4 | Coniferous | 2.7 |
|  | Low | 72.0 | Mixed | 5.5 |
| Foot slope | High | 4.9 | Broadleaf | 12.9 |
|  | Medium | 38.5 | Coniferous | 0.5 |
|  | Low | 56.6 | Mixed | 86.6 |
| Slope | High | 33.4 | Broadleaf | 4.5 |
|  | Medium | 46.3 | Coniferous | 20.1 |
|  | Low | 20.3 | Mixed | 75.4 |
| Ridge | High | 86.0 | Coniferous | 13.6 |
|  | Medium | 14.0 | Mixed | 86.4 |

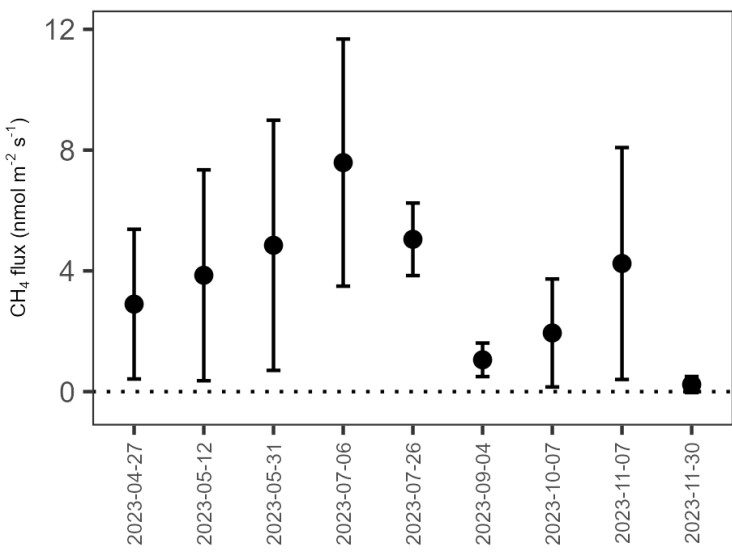

**Figure A1. Seasonal variation in soil CH$_4$ fluxes from wetlands (means and standard error, n = 3).**

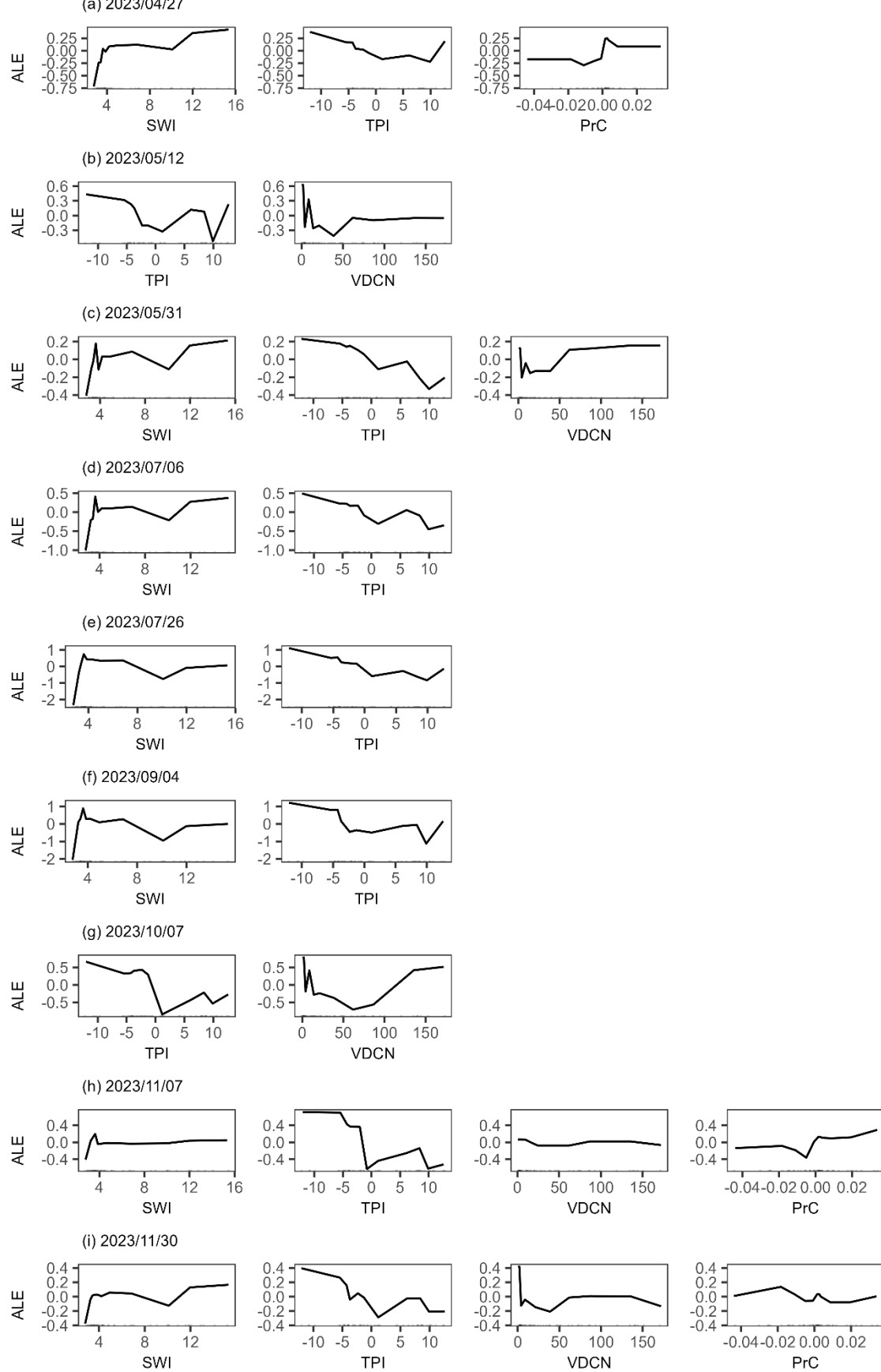

Figure A2. Accumulated local effect (ALE) plots for the quantile regression forest (QRF) models applied to non-waterlogged soil CH₄ fluxes at each measurement date.

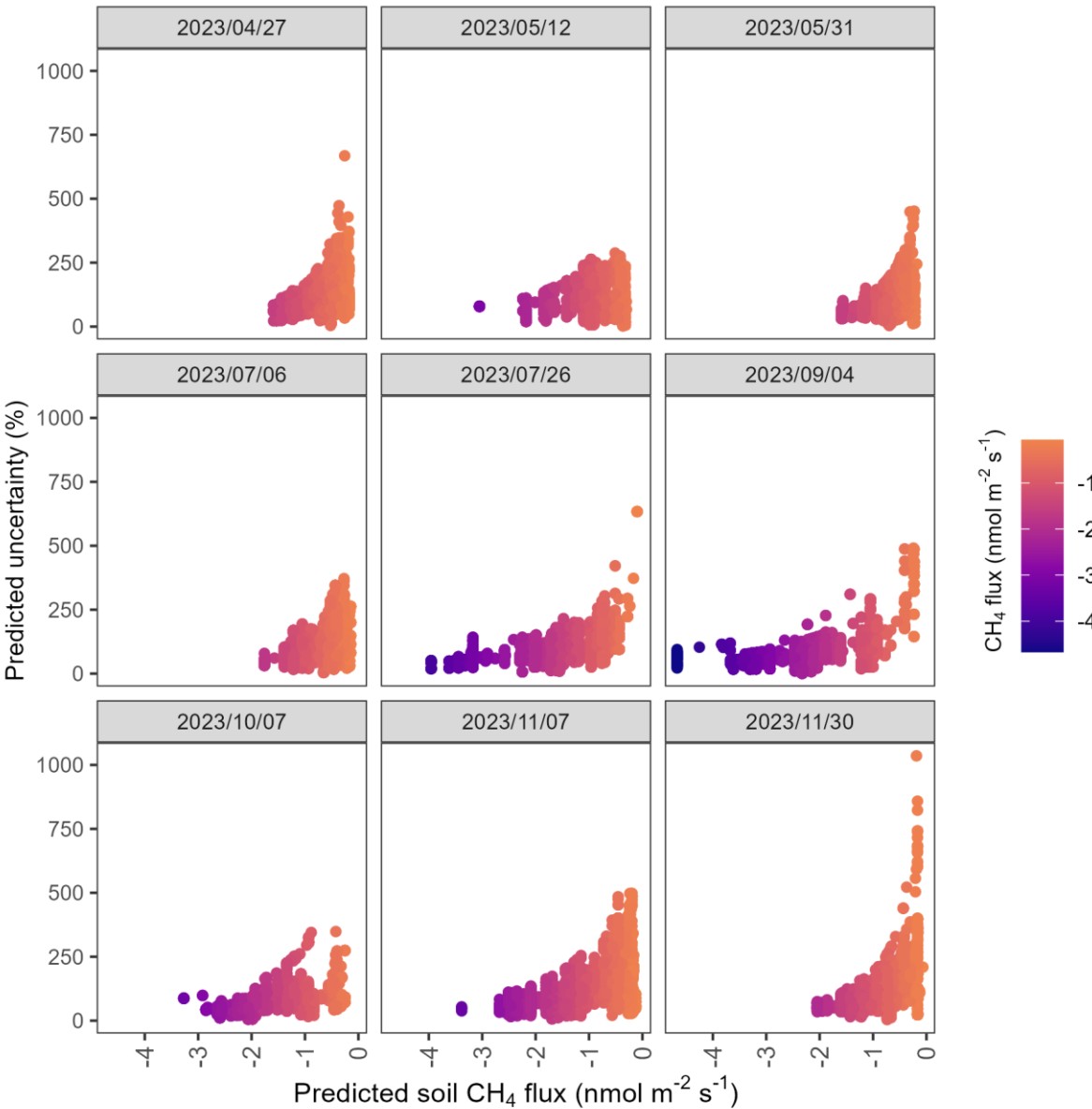

**Figure A3. Relationships between predicted uncertainty and predicted soil CH₄ fluxes using quantile regression forest (QRF) models applied to non-waterlogged soil CH₄ fluxes at each measurement date. The highest uncertainty is observed for a near-zero prediction.**

**Data availability**

The data used in this study are available at The Kyoto University Research Information Repository (KURENAI, DOI: doi.org/10.57723/kds605755)

**Author contribution**

DE had the original idea of this research. DE and SKP designed the research framework with suggestions from MD. SKP, DE, and KY conducted the vegetation survey and flux measurements. SKP analyzed and performed the modeling under the supervision of DE. SKP wrote the manuscript that was critically reviewed and edited by all co-authors.

**Competing interest**

The authors declare that they have no conflict of interest.

**Acknowledgement**

We would like to thank the staff of the Ashiu Forest Station of the Field Science Education and Research Centre, Kyoto University for enabling this research, providing the access to the forest, and for sharing the Digital Elevation Model (DEM) and climate data. We are grateful to Yusuke Onoda for sharing orthoimages of our study area, Takumi Mochidome for his helpful discussion about GIS and Lucie Bivaud and Makoto Nagasawa for their help during the field survey.

**Funding**

The research was supported by grants from the Research Institute for Sustainable Humanosphere (RISH), Kyoto University, the Japan Society for the Promotion of Science (KAKENHI Grant no. JP24K01797) and the SPRING fellowship program from Japan Science and Technology (JST SPRING Grant no. JPMJSP2110).

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
