# Peer review of "Upscaling of soil methane fluxes from topographic attributes derived from a digital elevation model in a cold temperate mountain forest"

_EGUsphere, 2025_

## Author Comment (AC1)

This file contains figures and tables that will be included or updated in the revised manuscript. We apologize but their format is not yet finalized

Table 1: Spearman-rank correlation between topographic and vegetation variables and measured soil water content, temperature, and chemistry (C, N, and pH)

|                   | Mean  | Mean soil   | C (%) | N (%) | рН    |
|-------------------|-------|-------------|-------|-------|-------|
|                   | SWC   | temperature |       |       |       |
| Aspect            | -0.14 | -0.10       | -0.02 | -0.07 | -0.06 |
| Profile curvature | -0.21 | 0.14        | 0.27  | 0.23  | -0.27 |
| Slope             | -0.17 | -0.24       | 0.25  | 0.22  | -0.51 |
| TPI 20m           | -0.35 | -0.11       | 0.45  | 0.39  | -0.34 |
| TPI 30m           | -0.31 | -0.08       | 0.64  | 0.55  | -0.46 |
| TPI 50m           | -0.33 | -0.08       | 0.59  | 0.50  | -0.42 |
| TWI saga          | 0.43  | 0.27        | -0.62 | -0.55 | 0.65  |
| VDCN 0.5ha        | -0.24 | -0.16       | 0.61  | 0.52  | -0.51 |
| VDCN 2.5ha        | -0.24 | -0.20       | 0.71  | 0.62  | -0.65 |
| VDCN 5ha          | -0.31 | -0.29       | 0.65  | 0.56  | -0.71 |
| CSA               | -0.37 | -0.29       | 0.22  | 0.13  | -0.25 |
| CONpc             | -0.17 | -0.56       | 0.30  | 0.19  | -0.41 |

Table: Selected variables for each measurement date, along with the R2 and root mean square error (RMSE) values to evaluate the accuracy of the quantile regression forests (QRFs) model. Importance scores of the selected variables are shown in parentheses, indicating their contribution to predicting soil CH4 fluxes. At the two dates where vegetation was selected, the model without and with vegetation (CSA) are compared (model with vegetation is in italic letters).

| Date       |               | Selected | variables  |            | $\mathbb{R}^2$ | RMSE |
|------------|---------------|----------|------------|------------|----------------|------|
|            | TWI_saga      | PrC      | TPI_30     |            |                |      |
| 2023/04/27 | (0.57)        | (0.58)   | (0.67)     |            | 0.53           | 0.52 |
| 2023/04/27 | TWI_saga      | CSA      | TPI_30     |            | 0.51           | 0.52 |
|            | (0.64)        | (0.42)   | (0.50)     |            |                |      |
| 2023/05/12 | TPI_30        | VDCN_5   |            |            | 0.31           | 0.82 |
| 2023/03/12 | (0.80)        | (0.66)   |            |            | 0.31           | 0.82 |
| 2023/05/31 | TWI_saga      | TPI_30   | VDCN_5     |            | 0.43           | 0.48 |
| 2023/03/31 | (0.55)        | (0.57)   | (0.42)     |            | 0.43           |      |
| 2023/07/06 | TWI_saga      | TPI_30   |            |            | 0.40           | 0.50 |
|            | (0.73)        | (0.60)   |            |            | 0.40           | 0.30 |
| 2022/07/27 | TWI_saga      | TPI_30   |            |            | 0.37           | 1.02 |
| 2023/07/26 | (0.80)        | (0.69)   |            |            | 0.57           | 1.02 |
| 2023/09/04 | TWI_saga      | TPI_30   |            | 0.40       |                | 1.18 |
| 2023/07/04 | (0.74)        | (0.85)   |            |            | 0.40           | 1.10 |
|            | TPI_30        | VDCN_5   |            |            |                |      |
| 2023/10/07 | (0.88)        | (0.58)   |            |            | 0.67           | 0.81 |
| 2023/10/07 | TPI_30 (0.77) | VDCN_5   | CSA (0.27) |            | 0.55           | 0.80 |
|            | 111_50 (0.77) | (0.39)   |            |            |                |      |
| 2023/11/07 | TWI_saga      | TPI_30   | VDCN_5     | PrC (0.45) | 0.59           | 0.66 |
| 2023/11/07 | (0.32)        | (0.84)   | (0.12)     | 110 (0.73) | 0.57           | 0.00 |

|            | TWI_saga | TPI_30 | VDCN_5 |            |      |      |
|------------|----------|--------|--------|------------|------|------|
| 2023/11/30 |          |        |        | PrC (0.28) | 0.51 | 0.56 |
|            | (0.32)   | (0.54) | (0.21) |            |      |      |

Table: Summary of the linear mixed model (LMMs) analyzing the relationship between the predicted soil CH4 fluxes and measured soil CH4 fluxes, where measurement periods were included as the random factor on both slope and intercept. The p-values of the fixed effect were for testing if the intercept was different from zero and the slope different from 1. The marginal (R2m) and conditional (R2c) coefficients of determination, and the root mean square error of the model are shown. Model without and with vegetation (CSA) are compared (model with vegetation is in italic letters).

| Fixed effe | ect: Predicted CH 4 flux |          | Random effect: measurement dates |            |           |         |
|------------|-------------------------------------|----------|----------------------------------|------------|-----------|---------|
| Estimate   |                                     |          | p-values                         |            | Intercept | Slope   |
| ± SE       |                                     | p-varues |                                  |            |           |         |
|            | $0.15 \pm 0.03$                     |          | 0.003                            |            | -0.05     | 0.03    |
| Intercept  | $[0.14 \pm 0.03]$                   | from 0   | [0.003]                          | 2023/04/27 | [-0.05]   | [0.02]  |
|            |                                     |          | 2.25×10 -9            |            |           |         |
| Slope      | $1.16 \pm 0.02$ $[1.16 \pm 0.02]$   | from 0   | [1.10×10 -            | 2023/05/12 | -0.02     | 0.02    |
|            |                                     |          | 8]]                              |            |           |         |
|            |                                     | from 1   | 0.92
[ 0.92 ]          | 2023/05/31 | -0.05     | 0.02    |
| $R^2m$     | 0.94 [0.94]                         |          |                                  | 2023/07/06 | -0.06     | 0.02    |
| $R^2c$     | 0.95 [0.95]                         |          |                                  | 2023/07/26 | 0.07      | 0.00    |
|            |                                     |          |                                  | 2023/09/04 | 0.14      | -0.03   |
|            |                                     |          |                                  | 2023/10/07 | 0.02      | -0.04   |
|            |                                     |          |                                  | 2023/10/0/ | [0.02]    | [-0.03] |
|            |                                     |          |                                  | 2023/11/07 | -0.01     | -0.04   |
|            |                                     |          |                                  | 2023/11/30 | -0.03     | 0.01    |

Figure: Comparison between predicted and measured CH4 fluxes aggregated within topographic position classes, vegetation type, and vegetation density. The 1:1 line is drawn.

Table: summary for Position

| Fixed effe    | ect: Predicted C | H 4 flux | Random effect: Positions |            |       |       |
|---------------|------------------|---------------------|--------------------------|------------|-------|-------|
| Estimate ± SE |                  | p values            |                          | Intercept  | Slope |       |
| Intercept     | $-0.17 \pm 0.04$ | from 0              | 0.006                    | Plain      | 0.01  | -0.02 |
| Slope         | $0.94\pm0.05$    | from 0              | 7.8E-05                  | Foot slope | 0.01  | 0.01  |
|               |                  | from 1              | 0.99                     | Slope      | -0.05 | 0.10  |
| $R^2m$        | 0.93             |                     |                          | Ridge      | 0.05  | -0.09 |
| $R^2c$        | 0.98             |                     |                          |            |       |       |
|               | 0.50             |                     |                          |            |       |       |

Table: summary for Vegetation type

| Fixed effect: Predicted CH 4 flux |                  |          |      | Random effect: Vegetation type |       |       |
|----------------------------------------------|------------------|----------|------|--------------------------------|-------|-------|
| Estimate ± SE                                |                  | p values |      | Intercept                      | Slope |       |
| Intercept                                    | $-0.19 \pm 0.07$ | from 0   | 0.08 | Broadleaf                      | 0.02  | 0.02  |
| Slope                                        | $0.82 \pm 0.09$  | from 0   | 0.01 | Coniferous                     | -0.09 | -0.14 |
|                                              |                  | from 1   | 0.99 | Mixed                          | 0.08  | 0.12  |
| $R^2m$                                       | 0.91             |          |      |                                |       |       |
| $R^2c$                                       | 0.96             |          |      |                                |       |       |
|                                              |                  |          |      |                                |       |       |

Table: summary for Vegetation density

| Fixed effect: Predicted CH 4 flux |                  |          |                       | Random effect: Vegetation density |       |       |
|----------------------------------------------|------------------|----------|-----------------------|-----------------------------------|-------|-------|
| Estimate ± SE                                |                  | p values |                       | Intercept                         | Slope |       |
| Intercept                                    | $-0.13 \pm 0.05$ | from 0   | 0.08                  | High                              | -0.01 | 0.02  |
| Slope                                        | $0.81 \pm 0.08$  | from 0   | 1.00×10 -2 | Medium                            | 0.06  | -0.14 |
|                                              |                  | from 1   | 0.99                  | Low                               | -0.05 | 0.11  |
| $R^2m$                                       | 0.80             |          |                       |                                   |       |       |
| $R^2c$                                       | 0.95             |          |                       |                                   |       |       |
|                                              |                  |          |                       |                                   |       |       |

Table 2.

Summary of the LMM analyzing the effects of topographic position, vegetation type and vegetation density on predicted soil CH4 fluxes. Pixel ID was included as a random effect, and spatial autocorrelation among residuals eliminated.  $\eta_p^2$  was calculated as the effect size of each explanatory variable

| Response variable                | Explanatory variables     | p-value | Effect size $(\eta_p^2)$ [with 95% CI] |
|----------------------------------|---------------------------|---------|----------------------------------------|
| Predicted CH 4 fluxes | Position [df=3]           |

Figure: ALE plots for all variables selected at each measurement date

Figure: Plot comparing the distribution of topographic variables between upland and wetland pixels

Table: Performance of model including wetland

| Fixed effect: Predicted                  |               |        |                       |            |       |       |  |  |
|------------------------------------------|---------------|--------|-----------------------|------------|-------|-------|--|--|
| CH 4 flux including           |               |        |                       |            |       |       |  |  |
| wetland Random effect: measurement dates |               |        |                       |            |       |       |  |  |
| Estimate ±                               |               |        |                       |            |       |       |  |  |
| SE                                       |               |        | p-values              |            |       |       |  |  |
| Intercept                                | $0.24\pm0.08$ | from 0 | 0.01                  | 2023/04/27 | 0.04  | 0.02  |  |  |
| Slope                                    | $1.15\pm0.05$ | from 0 | 2.05×10 -8 | 2023/05/12 | -0.14 | -0.08 |  |  |
|                                          |               | from 1 | 0.91                  | 2023/05/31 | 0.11  | 0.07  |  |  |
| $R^2m$                                   | 0.70          |        |                       | 2023/07/06 | -0.19 | -0.11 |  |  |
| $R^2c$                                   | 0.70          |        |                       | 2023/07/26 | 0.07  | 0.04  |  |  |
|                                          |               |        |                       | 2023/09/04 | 0.01  | 0.01  |  |  |
|                                          |               |        |                       | 2023/10/07 | 0.05  | 0.03  |  |  |
|                                          |               |        |                       | 2023/11/07 | 0.11  | 0.07  |  |  |
|                                          |               |        |                       | 2023/11/30 | -0.06 | -0.04 |  |  |

---

## Author Response (AR1)

Editor and Reviewer's comments

**Editor**

I have read your point-by-point rebuttal letter in response to the comments made by the two reviewers. I am happy for you to proceed and post the revised version of the manuscript. In your revision, identify the existing knowledge gaps that justify your machine learning approach to upscale soil CH4 fluxes across topographically complex forest landscapes. Justify all the methods used to test your hypothesis and address all the other issues raised by the reviewers, including the need for model validation; how the model deals with hydrologically heterogeneous landscape; variable choice; and scale mismatch in the validation approach.

We updated the last section of the introduction of our manuscript to better highlight the current knowledge gaps on which our objective and hypotheses are based. The comments of the two reviewers were useful to better justify the methods we used, our variable selection strategy, as well as the model validation, especially to deal with the scale mismatch issue. We sincerely thank them.

**Reviewer 1**

Overall Assessment

This manuscript presents a machine learning approach to upscale soil CH4 flux measurements across a topographically complex forest landscape using quantile regression forest models with topographic predictors. While the study addresses important questions about spatial controls on soil CH4 fluxes, there is room to improve methodologies to better differentiate between mechanistic and predictive statistics, and to contextualize the landscape-scale conclusions.

Major Comments:

1. The study assumes that topographic indices (TWI, TPI, VDCN) accurately represent soil moisture patterns that drive CH4 fluxes, but never measures soil moisture or temperature at sampling locations to validate this assumption. While the topographic predictors successfully predict CH4 fluxes, the mechanistic pathway (topography → soil moisture → CH4 flux) remains unverified. Without ground-truthing, it's unclear whether the correlations reflect the proposed moisture mechanisms or other covarying factors.

Please either acknowledge this limitation more explicitly or provide basic validation by measuring volumetric water content at a subset of locations to demonstrate that topographic predictors correlate with actual soil moisture conditions.

We measured soil water content (and temperature) at each collar at each CH4 flux measurements but did not judge useful to include these data, which was not sensible. We have added these measurements in the revised manuscript (**lines 212-213 and 329-301, Table 1, and Figure 2c,d**). In fact, prior to developing the model, we examined the Spearman-rank correlation between measured soil features and several topographic and vegetation variables. We found significant correlations between soil moisture and chemistry in one hand, and TPI, TWI, VDCN, PrC, basal area (BA) and conifer contribution to BA on the other. This supports the use of topographic and vegetation variables as effective predictors of CH4 fluxes in our landscape These correlations have been added to the manuscript (**lines 135-136 and Table A1**). The vegetation variables were not included in the final model because they did not improve model accuracy (**lines 343-344 and Table A3**).

However, incorporating soil moisture as an intermediate variable that would need to be scaled up to the landscape level would introduce an additional layer of uncertainty. Our strategy was to directly predict CH4 fluxes using topographic variables as proxies for soil moisture and related environmental gradients. We clarified this strategy in the revised manuscript (**lines 88-90**).

We nevertheless fully agree that no statistical method is mechanistic, and all can be biased by the existence of confounding factors. We have highlighted this limitation more explicitly in the revised manuscript (**lines 459-462**).

2. The study excludes wetlands from their predictive framework, leaving wetland pixels unmapped, but provides insufficient guidance on how their upland-only predictions should be applied to real forest landscapes. Most forests contain wet patches, seeps, or seasonally saturated areas that may not be classified as "wetlands" in standard remote sensing products but could function as significant CH4 sources. The authors' approach of simply excluding these areas creates uncertainty about how their upland flux predictions should be applied when: (a) wet patches exist but aren't formally classified as wetlands, (b) the boundary between "upland" and "wetland" conditions varies seasonally or with precipitation, and (c) their results are used to parameterize larger-scale models that need to handle mixed hydrologic conditions.

Provide clearer guidance on how to classify and handle hydrologically diverse areas when applying these results. Discuss what topographic or hydrologic thresholds define the boundaries of their "upland" predictions, and suggest approaches for handling wet patches that fall between clear upland and wetland classifications. This would help users appropriately

apply their upland flux relationships while avoiding systematic underestimation of emissions from hydrologically complex forest landscapes.

In our study, only permanent wetlands in the plain area were excluded (3 of 55 collars for data, less than 1% of the landscape pixels). Wet patches, which had temporarily water-saturated soils, were not excluded. Two collars were located in such wet patches, and positive fluxes were measured once on both collars. Therefore, to answer comment (a), areas with non-permanently saturated soil are included in the upland prediction and we clarified this point in the revised manuscript (**lines 233-234 and 304-305**). We also acknowledged that our random forest models did not predict median positive fluxes, but the possibility of positive fluxes is reflected in the large uncertainties associated with near-zero fluxes (**lines 495-497**).

Regarding (b), we acknowledged that using a fixed boundary between "upland" and "wetland" conditions, although these boundaries may vary seasonally depending on the balance between precipitation and evaporation, can increase uncertainties in $CH_4$ flux prediction. Predicting the temporal variations of these boundaries was beyond the scope of this work, and, at our site, wetlands represent only 1% of the pixels, and their boundaries even less. We have discussed the limitations of using static boundaries in more detail in the revised manuscript (**lines 166-170**).

Regarding (c), wetland exclusion, although acceptable in our 40-ha study area, where wetlands represent only 1% of the area, would overestimate CH4 uptake if incorrectly applied at larger scales, i.e., to the entire upper Yura River catchment in our case, for example, or to other hydrologically complex forest landscapes. We have already mentioned in our manuscript that sinks and sources should be modelled separately in the case of larger areas with mixed hydrological conditions. We have emphasized this point further in the revised discussion (**lines 484-486**).

For permanent wetland mapping, we collected additional GPS positions at the edges and within the three wetland patches. We then used SWI, profile curvature, slope, and VDCN to predict wetland locations. The method is detailed in a new supplementary file. Their boundaries were refined by visual inspection. A posteriori, pixel classified as wetland had TWI values above 8.1, profile curvature between -0.003 and 0.001, slope values below 6.8 for slope, and VDCN values below 2.2. It has been illustrated by a plot comparing the distribution of these topographic variables between upland and wetland pixels (**lines 162-165, 170-171 and Figure S1**).

3. Table A2 shows a significant three-way interaction (Position × Vegetation × Date, p = 0.04), yet the authors conclude that position and vegetation have no effects based on their lack of

selection in the later random forest models. This understates the importance of the interaction effects in their mechanistic descriptive modeling, even if it does not provide additional predictive power in the landscape scaling.

The authors should acknowledge that the significant interaction indicates vegetation and position effects are present but depend on specific combinations and timing. In the discussion of Jevon et al. (lines 372-382), note that while vegetation interactions were significant in the LMM, vegetation wasn't selected in RF models because continuous topographic variables captured relevant gradients more effectively for prediction.

We apologized but we should not have included interactions in the model, as some of them are missing. For example, there are no "pure" broadleaved areas on the ridge. Models with interactions would be rank-deficient. However, following the suggestion of the second reviewer, we significantly expanded the analysis of the influence of vegetation of CH4 fluxes, using not only the vegetation type but also its density. As landscape attributes, we now have (i) topographic position (plain, foot slope, slope, and ridge), (ii) vegetation type (broadleaf, coniferous, and mixed), and (iii) vegetation density (high, medium, and low). The effect sizes of topographic position, vegetation type, and vegetation density were 0.43, 0.006, and 0.11, respectively, highlighting the dominant role of topography over vegetation in the spatial variability of CH4 fluxes. The effect sizes have been added to the ANOVA table in the revised manuscript (**lines 269-270, 284-287, 376-377 and Table A6**).

4. The LMM results (Table A2) report only p-values without effect sizes, making it impossible to assess practical significance. Similarly, while RF variable importance scores are reported in a table, the magnitude and direction of predictor effects aren't clear in text/discussion.

Please rreport standardized coefficients for LMM factors to show effect magnitudes alongside statistical significance. For RF models, clarify the interpretation of relationships for key predictors (e.g., whether higher TPI increases or decreases CH4 uptake).

We recognise that failing to consider the effect sizes of variables used in linear models can lead to an underestimation of their potential mechanistic importance, even if these factors do not improve the performance of the random forest models. Effect size are now reported (**see our response to your previous comment and lines 376-377 and Table A6**).

For RF models, the direction of predictive effects is now illustrate using plots of accumulated local effects (ALE), which is effective if predictors are correlated with each other. The interpretation of the ALE analysis has been added to the revised

manuscript, as well as their graphical representation (**lines 249-256 and Figure A2**). In summary, for the two most influential predictors, low CH4 uptake rates were associated with high TWI values, while they were associated with low TPI values. We will discuss the magnitude and direction of predictive effects in more detail in the revised manuscript (**lines 340-342**).

5. Scale mismatch in validation approach The authors validate their predictions by comparing point measurements (20 cm diameter chambers) with pixel-level predictions (5m resolution), despite using predictors calculated at even coarser scales (e.g., 30m radius for TPI). This scale mismatch may actually understate the model's true predictive accuracy by forcing landscape-scale predictors to explain fine-scale chamber measurements that inevitably include local variability beyond what topographic indices can capture. The current validation approach tests whether coarse-resolution environmental variables can predict point-level flux heterogeneity, rather than testing the model's ability to capture the landscape-scale flux patterns it's designed to represent. Consider validating at aggregated scales that better match the conceptual basis of the predictors. Compare predicted vs. observed mean fluxes within topographic position classes (ridge/slope/foot slope/plain) or other meaningful landscape units to test whether the model captures the spatial patterns it's intended to represent. This approach would provide a more appropriate assessment of model performance for landscape-scale applications. Additionally, measuring soil moisture at chamber locations would help validate the mechanistic assumption that topographic predictors accurately represent the moisture conditions driving CH4 fluxes, allowing separation of prediction errors due to scale mismatch from errors due to invalid mechanistic assumptions.

To clarify, all predictors are calculated at pixel size (5 by 5 m), including TPI. For each 5 by 5 pixels, TPI is calculated based on the elevation of that pixel relative to the surrounding pixels within a radius of 20, 30, or 50 m. That being said, we fully agree that there is a scale mismatch between collar and pixel size. We appreciate your suggestion to validate at aggregate scales by comparing predicted and observed mean fluxes within topographic position classes. The $R^2m$ of linear mixed model between predicted and measured fluxes at the four topographic position for 9 measurement dates (n=36) was 0.93. We did this not only using topographic position classes, but also considering vegetation types and density classes. This validation has been included in the revised manuscript (**lines 362-365 , Tables A5 and Figures 4**) and the scale mismatch recognised in the discussion (**lines 465-471**).

As mentioned in the response to your first comment, we measured soil water content (and temperature) at each collar at each time CH4 flux measurements but did not judge useful

to include these data. We have added these measurements in the revised manuscript **(lines 299-301, Table 1 and Figure 2c,d)**.

Reviewer 2

The manuscript upscales soil methane fluxes with a digital terrain model in a forested landscape. It is nicely written, and the topic is relatively novel and worth investigating. However, there are certain issues that should be covered better.

1. The current analysis and methodology seems to have a double structure: there is a quantile regression forest analysis for upscaling methane fluxes for different dates and then there is a mix of different traditional statistical techniques (e.g., ANOVA, linear mixed models) for looking at relationships between different environmental characteristics (including topography and methane fluxes). I feel that this structure is a bit complex and some of the issues are done twice but with different methods. Therefore, I suggest simplifying the methodological approach and justifying better why certain analyses are conducted.

We appreciate the reviewer's observation regarding the complexity of our initial analytical framework. In the revised manuscript, we simplify the analytical approach. We first preselected terrain attributes (topography, vegetation) correlated with measured soil features. Next, we upscaled measured CH4 fluxes to the landscape level using a quantile regression forest (QRF) model and the preselected variables. Then, we tested whether predicted CH4 fluxes differed among landscape positions, vegetation types, and vegetation densities using a linear mixed model that accounted for spatial autocorrelation. Finally, we analysed the temporal variations of the upscaled fluxes. We believe that this streamlined approach reduces methodological redundancy and maintains consistency between scaling analyses and subsequent interpretations of spatial patterns. We have restructured the "methods" and "results" sections accordingly.

For instance, why is linear regression conducted between measured and predicted fluxes? Isn't it sufficient to provide observed-predicted plots?

We understand that providing observed–predicted plots may seem sufficient; however, we included linear regressions between measured and predicted fluxes to quantitatively assess the performance of the QRF models. This provided an objective

description of these plots, allowing us to test for the absence of bias, i.e., an intercept not significantly different from 0 and a slope not significantly different from 1. Furthermore, as the first reviewer pointed out a scale mismatch between collar and pixel sizes, we followed his suggestion and conducted additional analyses at aggregated scales by comparing predicted and observed fluxes aggregated by topographic position classes, vegetation density classes and vegetation type classes (**lines 362-365 , Tables A5 and Figures 4**).

Why is there a need to conduct separate linear regression between topography and methane fluxes in addition to quantile regression forests?

We agree with the reviewer that performing a separate linear regression between topography and measured CH4 fluxes was not justified, as the quantile regression forest model already captured nonlinear relationships. Therefore, we removed this analysis and instead present a correlation table among topographic variables and vegetation attributes in one hand, and soil moisture, temperature and chemistry on the other (**lines 133-135 and Table A1**). We are now reporting differences in predicted fluxes at the landscape level across topographic and vegetation classes, as mentioned above (**lines 362-365 , Tables A5 and Figures 4**). In other words, we deleted all statistical analyses done on measured CH4 fluxes (before QRF modelling) and strengthened the statistical analysis on predicted CH4 fluxes at landscape level (after QRF modelling).

2. In relation to the first point, there could also be additional analyses that have not been conducted. It is a bit unclear to me what is the logic in predicting temporally dynamic methane fluxes with temporally static topographic variables.

Why not to test also a model with both temporally static but spatially distributed topographic variables and temporally dynamic but spatially uniform climate/weather variables such as API (there could be possibilities for including also other weather-related variables)?

Because all pixels will have the same values for weather variables on a given date, it would be pointless to include them in RF models to predict spatial heterogeneity of landscape-scale fluxes. Previous works using similar RF modelling also run their models separately for each season without including weather data (e.g. Warner et al, 2019, Agric For Meteorol 264:80–91; Vainio et al, 2021, Biogeosciences 18:2003–2025). We justified our choice in the revised manuscript (**lines 262-266**).

Could the soil variables be included also in the quantile regression forest to test their strength in addition to the topographic variables?

Soil variables could potentially be included in the RF model but then the RF could not be used to predict flux at landscape level where only topographical and vegetation predictors

are available at pixel level. Of course, it would have been possible to use other RFs to upscale soil variables at the landscape level and use the predicted soil variables to predict methane fluxes, but it would add additional layers of uncertainties. We also justified this choice in the revised manuscript (**lines 88-90**).

Why is vegetation (or actually tree) information condensed into one categorical variable (forest type based on tree types? You could also have continuous variables about the tree species presence and abundance.

Thank you for this suggestion. We agree that we should have used a continuous variable for vegetation type in the model, and only used the categories for post hoc statistical tests. We also agree that it is useful to include vegetation density in addition to vegetation type. In the revised manuscript, in addition to topography, we tested two continuous variables in the quantile regression forest model: basal area for vegetation density and relative basal area of coniferous trees to basal area for vegetation type (**see lines 228-229**). The vegetation variables were not included in the final models because either they were not selected or did not improve model performance (**lines 343-344 and Table A3**). After QRF modelling, we used (i) topographic position (plain, foot slope, slope, and ridge), (ii) vegetation type (broadleaf, coniferous, and mixed), and (iii) vegetation density (high, medium, and low) to test the influence of these landscape attributes on the spatial variability of predicted CH4 fluxes (**lines 376-378**). The effect sizes of these three categorical variables have been added to the ANOVA table in the revised manuscript (Table A6).

3. The selection of the topographic variables for upscaling is relatively arbitrary. Why were these specific variables chosen and not others (listed e.g., in Ågren et al 2021, https://doi.org/10.1016/j.geoderma.2021.115280).

Many variables can be derived from DEM and selection is necessary to avoid overparameterization due to redundancies. Our preselection was motivated by the fact that methane fluxes are related to microbial activities (methanotrophic and methanogenic in our case), which are controlled by soil moisture and chemistry (C, N, pH), and, to a lesser extent, temperature. We examined the Spearman-rank correlation between measured soil water content, temperature and chemistry (C, N and pH) on the one hand, and several topographic and vegetation attributes on the other. We included the rational of our variable pre selection and added a table with the

correlation coefficient as an appendix in the revised manuscript (**lines 131-136 and Table A1**). Furthermore, it is unclear why SAGA wetness index was not used instead of the traditional topographic wetness index, as the SAGA version spreads high values in the flat areas.

We thank the reviewer for this insightful comment. In the revised analysis, we adopted the SAGA Wetness Index instead of the traditional Topographic Wetness Index (TWI). This is in **lines 131-132**. The reviewer's suggestion helped us recognize that the SAGA version provides a more accurate representation of wetness distribution, especially in flat areas, where the traditional TWI may direct the flow in the wrong directions, thereby distorting the flow accumulation.

Similarly, topographic position index should have been calculated with multiple neighborhood radiuses and vertical distance to streams with multiple stream networks. Now it is even unclear how the stream network was calculated and streams initiated when calculating the layer.

We acknowledge the reviewer's comment that multiple radii should be considered for TPI and multiple stream network for VDCN, as these variables are highly scale-dependent. TPI was calculated using neighborhood radii of 20, 30, and 50 m. To calculate VDCN, the DEM was first filled, and then flow accumulation layers were generated using the multiple flow direction method. The resulting flow accumulation raster was then used to create topographically defined flow channel networks, applying flow initiation thresholds of 0.5, 2.5, and 5 ha. VDCN was subsequently calculated for each threshold.

Finally, we chose the TPI with 30 m radii and VDCN with initiation thresholds of 5 ha, as they had the highest Spearman correlations with the soil variables, as explain above. However, the model performance in terms of its ability to predict measured fluxes were very similar when using any of combinations of these three TPI and VDCN. Detailed explanation and the correlation table have been added in the revised manuscript (**lines 150-152 and 159-160**).

4.Research hypotheses and result section are not well aligned with each other. Particularly, section 3.1 does not seem to address any of the hypotheses. I would suggest phrasing the hypotheses/research questions so that they are answered one by one in the results section.

We have a slightly different opinion here. For us, the research hypothesis should be well-aligned with the discussion, not with the result section, because hypotheses should be discussed based on the results but also additional information available from the literature. The result section follows a step-by-step organisation: the data, the model, the post hoc analysis.

Furthermore, also the methods section could be organized in the same way. Now result section starts with research for such methods that were described in the end of the methods section.

Thank you, here we agree with you that methods and results should be better aligned. The structure of these sections has been improved, to better align the method section with the updated result section.

5. Novelty value of the research is not entirely clear yet. Is the main novelty about analyzing the role of topography on methane fluxes at different times of snow-free season? If yes, this could be highlighted more in the introduction and also in the conclusions section.

Yes, our objective was to analyse the role of terrain attributes (topography, vegetation) on methane fluxes throughout the snow-free season in a topographically complex mountainous landscape, and how the spatial heterogeneity of predicted flux at the landscape level vary over time. We have stated this in the introduction (**lines 80-85**) and also in the conclusion (**lines 574-575**).

More detailed/minor comments:

- l14: "aimed to investigate" -> "investigated"; i.e., you can use stronger language

Thank you, we updated the sentence.

- l79: should it be "have been" instead of "were" to be more consistent with tenses. Also otherwise, it is best to write the introduction in present tense.

We have replaced "were" by "have been" as suggested. This may be an old-fashioned way of writing, but for us, the introduction should be in the present tense to describe the context or state general knowledge, while the past tense is used for specific knowledge related to previous experiences and our own work.

- l82: can the km2 be written in ha so that same unit is used for all referenced studies

We have updated the units

- l85: Can you start just by writing "We assess". Overall, it would be best if you would use active voice throughout. Now, you use partly passive and partly active voice in the methods section.

We have adopted active voice wherever it was possible

- Figure 1: Can you also show the location of the area within Japan/Honshu?

A map of Japan has been added to Figure 1

- l17: how were the coverages for the different land cover types estimated?

We have extended this section in the revised manuscript (**lines 162-176**). To distinguish wetland and upland areas, we collected additional GPS positions at the edges and within each wetland, in addition to the positions of the 55 sampling points. We then used SWI, PrC, slope, and VDCN to predict the locations of wetlands using a machine learning approach now described in the supplementary file (Note S1) . Finally, the boundaries between wetlands and

uplands were refined by visual inspection. For river mapping, pixels corresponding to rivers were identified in the channel network raster, which was calculated using a 5-ha initiation threshold. Slope angle and TPI at 30 m radius were used to partition the upland into ridges, slopes, foot slopes, and the plain. Locations with TPI values of 5 or greater were defined as ridges, representing locally elevated, convex surfaces. Locations with TPI values ≤ -5 were defined as foot slopes, representing concave or lower landscape positions. Areas with intermediate TPI values (−5 < TPI < 5) were further divided according to slope angle: sites with slope > 18° were defined as slopes, and those with slope ≤ 18° were defined as plains. This has been added in the revised manuscript (**lines 162-179**).

- How was the measurement point sampling designed? Purposeful sampling or somehow randomized designed? How were the wetland measurement points sampled? Did you use boardwalks when measuring methane fluxes from wetlands?

The sampling points were chosen along three transects perpendicular to the main river, from the plain to the ridges covering two slopes (south-facing and north-facing), as well as in a lateral canyon, and along transects parallel to the main river, on the plain, above the foot slope and on a ridge. The sampling was designed to encompass the landscape heterogeneity, therefore purposeful, while being constrained by the geography of the site and safety considerations. The wetland patches were small enough that boardwalks were not required. When measuring fluxes in wetland, we took care to avoid trampling the soil near the collars, taking advantage of the abundant presence of stones and coarse woody debris. These additional information have been added in the revised manuscript (**lines 113-116**).

- l145/147: maybe better to use "spatial resolution", "pixel size" or "grid" instead of "mesh"

  We replaced mesh by grid

- l148: you write in parentheses both "less than" and "≤". Either or is sufficient. Actually, it should be "less than or equal to".

We removed "less than"

- l161: What method was used to fill the DEM?

We used the Wang and Liu (2006) method, implemented by built-in DEM filling option in QGIS. We added the reference (**lines 145-146**)

- Did you upscale the vegetation/tree classification for the whole study area?

Yes, we upscaled basal area of conifers and broadleaved trees separately for the entire study area using SWI, TPI, VDCN, and the normalized vegetation index (NDVI) from of the 55

10-meter radius census plots (**lines 182-192**). We used a machine learning approach now described in the supplementary file (Note S2).

- l195: you can delete "In this study". It is self-evident that you are describing "this study"

We deleted it.

- l203: How were the vegetation types used as predictors? One categorical predictor with three different values? Why not to use continuous predictors related to vegetation and soil?

We thank you for this suggestion. We re-evaluated the RF models using two continuous predictor variables: vegetation type (relative basal area of coniferous trees to basal area) and vegetation density (basal area) (**lines 227-229**). Vegetation type was never selected while vegetation density (basal area) was selected twice, in April and October. However, it turns out that excluding basal area from the initial list of variables increased the model performance on both dates. We therefore did not include basal area in the final models used to upscale fluxes to the landscape level, thus avoiding adding an additional layer of uncertainty. We added a comparative table in the appendix A3 (see also in the 'results' and 'discussion' section, (**lines 342-343, 423-424**)

We continue to use categorical variables for post hoc statistical tests. For vegetation type and vegetation density, the distribution of the variable was divided into three categories: above the upper quartile, within the interquartile range, and below the lower upper quartile. As previously mentioned, soil variables were not included in the RF model because they are not available at pixel level. We could have used other RF models to upscale the soil variables to the landscape level, but this would have added additional layer of uncertainties. Soil variables are well correlated with topographical and vegetation variables, as previously mentioned. Therefore, we consider them to be indirectly included in the final RF models.

- VSURF: did you employ all three steps of the method?

Yes, we follow the method of Genuer et al. (2010) for variable selection. We provided more details about the different steps of variable selection in the revised manuscript (**lines 234-239**).

- l217: Why did you use a separate package for variable importance? They can be obtained from random forest directly. What metric was used to assess the importance?

Although variable importance can be obtained directly from the random forest algorithm, we calculated variable importance using the vip package (Greenwell and Boehmke, 2020). Variable importance scores were estimated using a permutation-based approach, in which the values of each predictor in the training data were randomly permuted to assess the

resulting change in model performance, as quantified by the adjusted R-squared value. A greater reduction in adjusted R2 indicated a higher importance of the predictor variable. We provided more details in the revised manuscript (**lines 246-249**).

- l235: How about temporal autocorrelation in the models?

We included pixel IDs as a random effect in the linear mixed model to account for repeated prediction at the same location, as mentioned **lines 280-283**.

- l242: What does "scaled" mean here?

For clarity, we have replaced "scaled soil CH4 fluxes" with "landscape-scale predicted soil CH4 fluxes"

- l288: How were the importance scores quantified?

As previously mentioned, importance scores were quantified using a permutation-based approach developed by Greenwell and Boehmke (2020) and implemented in the VIF package (**see lines 246-249**).

- Figure 3: Is the line 1:1-line?

Yes, we revised the caption of this figure.

- Figure 5: Can you have the measured fluxes in the same figure?

We modified Figure 5 by reporting the average of all flux measurements during the snow-free season aggregated by topographic position, vegetation density, and forest type. We added the average measured fluxes, as suggested (Figure 6 in the revised manuscript).

- l458: Can you provide some results about the models including wetland points in the supplementary material? Now this feels like speculation.

A comparison of model performance with and without wetland measurements has been included in an appendix (Table A2).

- l516: You did not really quantify the dominant role of topography as your quantile regression models had mostly just topography predictors.

We agree that only topographical predictors were used in the previous data analysis. Because we included vegetation in the post hoc statistical test, we can now safely conclude, based on the effect size, on the dominant role of topography on the spatial variation of soil CH4 fluxes (Table A6).

---

## Author Response (AR2)

Dear Dr Erika Buscaro,

We are pleased to submit the corrected version of our manuscript.

We answer to your comments and the comments of Referee #2 below.

Sincerely

Sumonta Kumar Paul

Editor comments

I have made some corrections and comments in the attached document. I suggest that you thoroughly revise the language and the use of the terminology 'upland' vs. 'wetland'.

Thank you for the time you spent on our manuscript. We proofread the manuscript again to fix language issues as much as we could.

We were also not satisfied with the terminology "upland". We agree with your suggestion: "non-waterlogged" and made all appropriate changes, or deleted "upland", depending on the context.

Basal area is a very common metrics for vegetation density in forest because, unlike stems of grass and herbaceous dicots that have quite similar diameter, the diameter of trees, particularly in unmanaged forests, shows a large range of diameter variation (from few centimeters to more than 1 meter in our case). Basal area reflects better land coverage and resource use than the number of trees.

Please also revise all figure legends and table captions to be stand-alone so that they can be understood without recourse to the main text (What? Where? Why?).

We have added details to the captions of the figures and tables.

Referee #2

l18-19: this sentence reads a bit awkwardly in the middle of results in the abstract. Please consider revising the sentence.

We deleted this sentence

l88-90: this sentence is out of place here. Consider deleting or revising.

We moved this sentence to the materials and methods section

l117: "accuracy less than": is this written correctly? Should it be "accuracy higher than"?

Yes, we edited the sentence "accurate to a radius of 5 m or less"

l131: saga should be SAGA

Corrected

l258: how did you calculate R2? There are multiple ways for calculating it.

$R^2$ was calculated as the square of the correlation between observed and cross-validated predicted fluxes, as implemented in the "caret" package (lines 248-250)

l264-266: please delete the sentence. Inclusion of spatially uniform variables would make sense only if you would include all the measurement dates into one model.

We deleted this sentence

l423-424: this does not read well in the beginning of discussion. Please revise the start of the section with a more general summary of results.

We moved this sentence to the end of the sub-section

l428: in this line, you first write SWI and then TWI. Should you use only SWI?

Thank you, we corrected the mistake

l466: overestimated->overestimate

Corrected

l470: was->were

Corrected

l552 and 580: landscape-scaled->landscape-scale

Corrected everywhere

In addition to these corrections, please make another round of proof-reading as there seem to be some small grammatical or other errors here and there.

We proofread the manuscript again to fix language issues as much as we could.